**Report**

**EMBO** *reports*

# Dynamic association of H3K36me3 with pericentromeric heterochromatin regulates its replication time

Sunil Kumar Pradhan [ID], Hui Zhang, Ksenia G Kolobynina, Alexander Rapp [ID], Maria Arroyo & M Cristina Cardoso [ID] [✉]

## Abstract

**The flexibility of the spatio-temporal genome replication program during development and disease highlights the regulatory role of plastic epigenetic mechanisms over genetic determinants. Histone post-translational modifications are broadly implicated in replication timing control, yet the specific mechanisms through which individual histone marks influence replication dynamics, particularly in heterochromatin, remain unclear. Here, we demonstrate that H3K36me3 dynamically enriches at pericentromeric heterochromatin, composed of major satellite DNA repeats, prior to replication during mid S phase in mouse embryonic stem cells. By knocking down lysine 36-specific methyltransferases or by targeting the H3K36M oncohistone to pericentromeric heterochromatin, we reduce global or local H3K36me3 levels, respectively, revealing its essential role in preserving the replication timing of constitutive heterochromatin. Loss of H3K36me3 accompanies increased RNA polymerase II serine-5 phosphorylation and lowered major satellite RNA levels, indicating transcriptional dysregulation. Notably, we identify a strand-specific contribution of major satellite forward transcripts in regulating the replication timing of constitutive heterochromatin and maintaining chromatin stability, highlighting the importance of non-coding RNAs as critical regulators of replication timing.**

**Keywords** H3K36me3; Pericentromeric Heterochromatin; Oncohistone; MajSat RNA; Replication Timing
**Subject Categories** Chromatin, Transcription & Genomics; DNA Replication, Recombination & Repair; Post-translational Modifications & Proteolysis

## Introduction

Chromatin duplication prior to cell division occurs in the S phase and progresses in a tightly regulated spatiotemporal pattern (Chagin et al, 2010). This is intrinsically related to the temporal order of coordinated firing of replication origins (replication timing) at distinct chromatin regions (Rhind and Gilbert, 2013). The precise coordination of genome replication timing is fundamental for preserving (epi)genome integrity during cellular proliferation (Alabert and Groth, 2012). The genome replication program displays remarkable flexibility through developmental stages, suggesting regulation beyond genetic elements (Marchal et al, 2019). Epigenetic mechanisms, particularly histone post-translational modifications (PTMs), have emerged as central regulators of replication timing by shaping chromatin accessibility and organization. Fluctuations in the histone acetylation levels directly reprogram replication timing in heterochromatin (Casas-Delucchi et al, 2012b, 2011). Replication timing also influences the epigenome and plays a key role in maintaining the epigenome. Changes in the global replication timing or that of constitutive heterochromatin leads to erroneous epigenetic inheritance (Heinz et al, 2018; Klein et al, 2021). However, the specific contribution of individual histone marks to the temporal control of DNA replication, particularly in heterochromatin, remains poorly understood.

Heterochromatin, characterized by its condensed chromatin structure of repetitive sequences, poses unique challenges during chromatin duplication to the replisome machinery and factors involved in chromatin reestablishment. Mouse pericentromeric heterochromatin (PCH) consists of tandem repeats of major satellite (MajSat) sequences, which are essential for maintaining genome stability (Jagannathan et al, 2018; Vissel and Choo, 1989). These highly condensed structures are enriched in heterochromatin marks such as H3K9me3 and H4K20me3, which recruit various heterochromatin-associated proteins to maintain the condensed state (Schotta et al, 2004). Despite their highly condensed nature, the PCH are not transcriptionally inert and are known to be expressed in a cell cycle and developmental stage-dependent manner (Lu and Gilbert, 2007; Probst et al, 2010). MajSat RNAs play crucial roles in PCH formation during early development and in maintaining heterochromatin condensation by recruiting and retaining heterochromatin-associated factors (Probst et al, 2010; Velazquez Camacho et al, 2017; Novo et al, 2022). However, the mechanisms behind the dynamics of MajSat RNA, the factors regulating these fluctuations, and their implication in chromatin organization, especially during chromatin duplication, remain elusive.

Cell Biology and Epigenetics, Department of Biology, Technical University of Darmstadt, 64287 Darmstadt, Germany. ✉E-mail: cardoso@bio.tu-darmstadt.de

In addition to other heterochromatin marks such as H3K9me3 and H4K20me3, the H3K36me3 mark has also been observed to be associated with the constitutive heterochromatin and, in some instances, to cooperate with H3K9me3, forming dual heterochromatin domains despite being also an open chromatin mark (Chantalat et al, 2011; Barral et al, 2022; Zhang et al, 2015). Typically, the presence of H3K4me3 and H3K36me3 at promoters and gene bodies, respectively, marks active gene expression (Ruthenburg et al, 2007; Huang and Zhu, 2018). While the association of H3K36me3 with transcription elongation in open chromatin is well understood, its role in heterochromatin is unclear.

Here, we elucidate the dynamic localization of H3K36me3 with PCH correlating with the genome replication progression and its role in regulating PCH replication timing in mouse embryonic stem cells. By reducing global H3K36me3 levels through knockdown of lysine methyltransferases and locally disrupting H3K36me3 by replacing the canonical histone H3.1 with oncohistone H3.1 K36M at PCH, we demonstrated its critical role in regulating heterochromatin replication timing and the transcriptional state of MajSat repeats. We further identified a strand-specific role of MajSat RNA in controlling the replication timing of PCH, suggesting non-coding RNA as a player in replication timing regulation. Overall, our findings reveal a heterochromatin-specific role of H3K36me3 and illustrate how multiple epigenetic factors coordinate to regulate DNA-dependent processes.

## Results and discussion

### Dynamic enrichment of H3K36me3 at pericentromeric heterochromatin prior to DNA replication

Initially, we sought to find possible histone post-translational modifications (PTMs) that might play regulatory roles in various aspects of the S phase in mouse embryonic stem cells (mESC J1), as histone PTMs are crucial for modulating chromatin structure and influencing gene expression during DNA replication. We took advantage of high-throughput imaging of asynchronously growing cell populations immunostained with DAPI, EdU, and prominent histone PTMs. We utilized DAPI as a proxy for DNA content and EdU to mark (non)replicating cells, allowing us to determine cell cycle stages of individual cells. Upon plotting the total DAPI intensity against the mean EdU intensity, we observed three clusters of cell populations (Sasaki et al, 1986). The cluster with lower DAPI and EdU intensities corresponds to G1, while the cluster with higher DAPI and lower EdU intensities represents G2. The population with higher EdU intensities represents replicating cells in S phase (Fig. EV1A). We found a large fraction of the cell population in S phase (close to 70%), validating the short G1 duration in mESCs (Coronado et al, 2013). We measured the sum intensities of the histone PTMs in different cell cycle stages, normalized to the DNA content, and further normalized to the G1 level to estimate the fold change in the histone PTM levels in the S phase and G2. We observed significant changes in the overall histone PTM levels as the cells entered the S phase (Fig. EV1B). There was an increase in histone acetylation marks associated with nascent chromatin (e.g., H4K5ac, H4K12ac, and H3K56ac) and active transcription marks (e.g., H3K4me3, H3K36me3, H3K27ac,

and H3K9ac) (Stewart-Morgan et al, 2020; Smolle and Workman, 2013). On the contrary, heterochromatin marks such as H3K27me3, H4K20me3, and H3K9me3 showed a significant drop in PTM levels in the S phase population. When analyzing the PTM levels in G2, we found that only a subset of these modifications returned to G1-like levels, suggesting incomplete reestablishment. Prominently, modifications like H3K9me2, H3K36me3, H4K8ac, and H4K12ac decreased further in G2 (summarized in Fig. EV1C). Among these, the levels of H4K12ac and H3K36me3 were also higher in the S phase. The H4K12ac, along with H4K5ac, is recruited to the nascent chromatin and gets removed in the process of chromatin maturation, explaining their cell cycle dynamics (Taddei et al, 1999; Alabert et al, 2014). However, the mechanism underlying the loss of H3K36me3 levels in G2 after an increase in S phase was unclear. The methylation of H3K27 and H3K36 follows a domain model of epigenetic reestablishment using a read-write mechanism (Alabert et al, 2020). While we observed a significant increase in the H3K27me3 levels in G2, suggesting its inheritance, we observed that H3K36me3 levels were not reestablished. Hence, we reasoned that H3K36me3 might have a role during chromatin duplication.

We further used high-resolution microscopy and statistical image analysis to infer the subnuclear distribution of H3K36me3 and correlated it with the cell cycle progression. We used the same approach as before, but here, we differentiate among the S phase stages to segregate the S phase population into S I, S II, and S III, based on the spatial pattern of the replication foci (RFi), along with G1 and G2. The functional nuclear architecture comprises the active nuclear compartment (ANC) and the inactive nuclear compartment (INC). The INC is enriched in repressive histone marks, while the ANC encompasses transcriptionally active regions (Cremer et al, 2015). Combined with high-resolution microscopy, we used the image analysis tool 'nucim' available on the statistical analysis platform R to segregate individual voxels within the nucleus into seven different chromatin compaction classes, using a hidden Markov random field model based on DNA intensity (see methods image analysis) (Fig. EV2A) (Schmid et al, 2017; Smeets et al, 2014). Using an intensity-weighted threshold method, we segmented and mapped replication foci (RFi) to individual compaction classes using cells in different S phase substages (Fig. 1A,B). While most of the RFi in the S I and S III were mapped to the ANC, a significant fraction of the RFi in S II mapped to the INC representing the constitutive heterochromatin (Fig. 1B). This validates the earlier observation of advanced replication timing of pericentromeric heterochromatin (PCH) comprised of the tandem repeats of major satellites (MajSat) in the S II phase in mESCs (Rausch et al, 2020; Smeets et al, 2014). We further mapped the histone modification to chromatin compaction classes to infer if it exhibits a dynamic association with different chromatin states (Pradhan and Cardoso, 2023). Mapping of H3K36me3 to chromatin compaction classes depending on cell cycle stages revealed its dynamic subnuclear distribution. We observed its enrichment with the most compacted chromatin classes in G1, S I, and S II and afterward dissociation (Fig. 1C). We next used a 3D image analysis approach as described earlier with minor modifications (Fig. EV2B,C) (Ollion et al, 2013; Pradhan et al, 2024). We directly segmented PCH (DAPI-dense regions) using a threshold- and object size- based approach and measured H3K36me3 levels normalized to the DNA in the corresponding cell cycle stages (Fig.

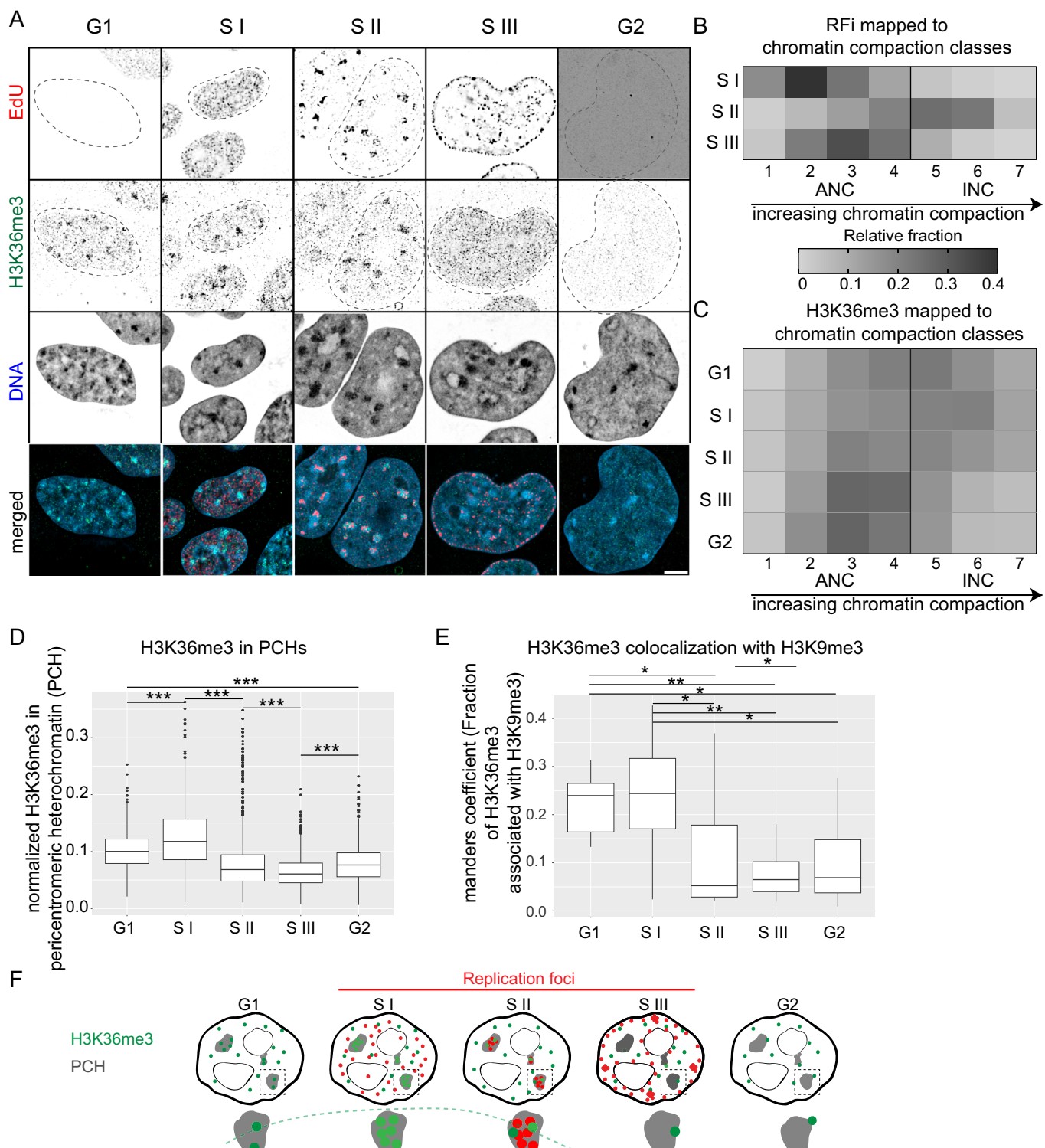

1D). As a validation, we also measured the relative association of H3K36me3 with the H3K9me3, which is known to be constitutively present in the PCH (Fig. 1E) (Martens et al, 2005). In both cases, we observed a relatively stronger enrichment of H3K36me3 in PCH in G1 and S I, with the highest in S I prior to the replication of PCH in S II. The relative enrichment of H3K36me3 in PCH dropped

already in S II and did not increase significantly afterward in S III or G2, suggesting the PTM is not reestablished post-replication.

These findings highlight the dynamic regulation of H3K36me3 during the S phase, with its transient enrichment in pericentric heterochromatin prior to the replication of these genomic regions in S II (Fig. 1F). This enrichment suggests a potential role in

Figure 1.  **H3K36me3 dynamically localizes to the pericentromeric heterochromatin prior to its replication.**

(A) Representative images show the RFi patterns [EdU (red) only in S phases], respective H3K36me3 (green), DNA (blue) and merged images. Scale bar: 5 µm. (B) Heatmap shows the RFi mapped to chromatin compaction classes (class 1, 2, 3, and 4- represent active nuclear compartment (ANC)), classes 5, 6, and 7- represent inactive nuclear compartment (INC)) in sub-S phase stages. (C) Heatmap shows the H3K36me3 mapped to chromatin compaction classes in cell cycle and sub-S phase stages. (D) The sum intensity of H3K36me3 (normalized to DNA sum intensity) in DAPI-rich pericentromeric heterochromatin was measured by segmenting the PCH and plotted over cell cycle stages, revealing an increased amount of H3K36me3 in G1, and S I. $n = 109$ cells (from three biological replicates) with at least ten cells from each cell cycle stage. Relevant significant differences (inferred from pairwise Wilcoxon test) are marked by asterisks (*), and all $p$ values are provided in the source data files. Relevant $p$ values from left to right are: G1-S I:$p = $1.3e-10, S I- S II: $p = $1.448134e-77, S II- S III:$p = $1.681883e-05, S III-G2:$p = $3.334893e-12, G1-G2:$p = $1.668588e-19. (E) Colocalization analysis between H3K9me3 and H3K36me3 using Mander's colocalization analysis reveals a similar trend. $n = 109$ cells (from three biological replicates) with at least ten cells from each cell cycle stage. Significant differences (inferred from pairwise Wilcoxon test) are marked by asterisks (*), and all $p$ values are provided in the source data files. Relevant $p$ values are: G1-S I: $p = 0.763183732$, G1-S II: $p = 0.0447557$, G1- S III: $p = 0.0053583$, G1- G2: $p = 0.0250435$, S I-S II: $p = 0.0364259$, S I-S III: $p = 0.0019442$, S I- G2: $p = 0.0213586$. For both panels (D, E), boxplots show the median (line), interquartile range (box: 25th to 75th percentile), and whiskers extending to data within 1.5× the IQR. n.s. not significant; *$p < 0.05$; **$p < 0.005$; ***$p < 0.0005$. (F) The illustration shows the dynamic localization of H3K36me3 to the pericentromeric heterochromatin marked with H3K9me3 in G1 and S I, and the relative enrichment drops in S II, and the mark does not reestablish in later stages. Source data are available online for this figure.

preparing PCH for replication, which is further supported by its dissociation post-replicative and the lack of reestablishment in G2. These observations, combined with the broader cell cycle dynamics of histone PTMs, offer insights into how histone modifications orchestrate chromatin state transitions essential for accurate replication and cell cycle progression.

## Reduced level of H3K36me3 delays the replication timing of pericentromeric heterochromatin

To gauge the possible role of the dynamic subnuclear distribution of H3K36me3 on chromatin duplication, we aimed to deplete the H3K36me3 modification specifically. The histone modification H3K36me3 is known to be an active transcription mark that predominantly gets modified by set2/SETD2 recruited by Ser2-phosphorylated RNA Pol II to the elongating transcription sites to maintain transcription fidelity, a process critical for ensuring precise gene expression and preventing errors during transcription (Venkatesh and Workman, 2013; Edmunds et al, 2008). Hence, its presence in the constitutive heterochromatin, known to be transcriptionally incompetent, poses questions about its role in heterochromatin. We aimed to lower the level of H3K36me3 to infer its potential roles in heterochromatin. Previous attempts to reduce H3K36me3 from the chromatin domains using knockdown of individual lysine methyltransferases (KMT) revealed none of the KMTs were solely responsible for the trimethylation of H3K36 in pericentromeric heterochromatin, with SETD2 having a substantial effect compared to the other KMTs such as NSDs, underscoring the complexity and potential redundancy in the regulation of H3K36me3 levels in different chromatin context (Chantalat et al, 2011). A similar trend was observed by knocking out SETD2 in mouse embryonic stem cells, which resulted in a minimal reduction of H3K36me3 levels in the dual domains of heterochromatin with H3K36me3/H3K9me3 (Barral et al, 2022). Hence, we used esiRNA against two major KMTs (NSD1 and SETD2) and eGFP (as a negative control) to reduce H3K36me3 and characterize its role in heterochromatin. We first validated the knockdown effect on individual KMTs using immunostaining and high-throughput microscopy and found a decreased level 36 h post-transfection for both KMTs (Fig. EV3A,B). We validated reduced levels of H3K36me3 at this time point by western blot using acid-extracted histones from both samples (Fig. EV3C). Additional functional characterization revealed a decrease in the S phase population upon KMTs knockdown, consistent with earlier

findings (Fig. EV3D) (Pai et al, 2017). We, then, used a pulse-chase approach to dissect the effect of H3K36me3 reduction on the replication program, focusing specifically on the PCH (Fig. EV3E). Briefly, a short pulse of EdU was followed by a chase with an excess of thymidine to block further EdU incorporation for 3 h, and cells were fixed. Both EdU and PCNA were detected, providing two snapshots in time of the RFi spatial patterns from which the replication progression was inferred. The S I spatial pattern was inferred from a negative EdU but positive PCNA signal, whereas cells with EdU signal were scored as being in S III when punctuated PCNA RFi was no longer detectable. The S II spatial pattern was inferred from the PCNA pattern, with EdU showing an S I pattern. These spatial patterns were specific to each experimental condition and were used for further analysis. Along with capturing the spatiotemporal replication dynamics, this approach provided a significant advantage over the Repli-seq method, as sequencing-based approaches often fail to cover tandem repetitive sequences such as MajSat repeats effectively. In controls (esiGFP treated cells), we did not observe any deviation of replication progression, and it was similar to the earlier observed replication program in mESCs, where euchromatin replication in S I was followed by the PCH in the S II, and nuclear periphery and nucleolar-associated chromatin replicated in S III (Rausch et al, 2020). Following knockdown of KMTs (esiKMT), we observed an aberration in the progression of the heterochromatin replication timing (Fig. EV3F). After the euchromatin replication in S I, we observed a concomitant replication of PCH with nuclear periphery and nucleolar-associated chromatin, whose replication timing is usually temporally separated.

The knockdown of KMTs effectively reduced the level of H3K36me3, and we could observe changes in the spatiotemporal replication dynamics, pointing towards H3K36me3 modulating the replication timing of PCH. To validate this causality, we intended to reduce the H3K36me3 locally at the PCH. The oncohistone H3.3 K36M has been shown to act as a dominant-negative mutant and is known to reduce H3K36me3 without affecting other methylations (Lewis et al, 2013). We achieved the reduction of H3K36me3 from PCH by taking advantage of the GFP nanotrap-mediated targeting strategy (Heinz and Cardoso, 2019; Rothbauer et al, 2006; Lindhout et al, 2007). In mESCs, we targeted EGFP-tagged H3.1 with or without the lysine 36 to methionine (K36M) mutation, a dominant-negative mutation that inhibits H3K36 methylation, to PCH by cotransfection with a GFP-binding protein (GBP) tagged MajSat polydactyl zinc finger (PZF) protein (GBP-MajSat). MajSat PZF

specifically binds to the major satellite DNA tandem repeats, and the GBP domain interacts with GFP, thus recruiting EGFP-tagged H3.1 to the PCH (Fig. EV4A). The incorporation of H3.1 into the histone octamer is replication-dependent, while the H3.3 incorporation is not coupled to the DNA synthesis (Tagami et al, 2004). Furthermore, the pericentromeric heterochromatin is enriched with H3.1 rather than H3.3, and replacing H3.1 with H3.3 leads to altered heterochromatin properties (Arfè et al, 2024). Hence, the strategy of targeting H3.1 to the PCH with or without the K36M mutation provides a unique approach to specifically reducing the H3K36me3 level in the PCH and investigating its direct role in replication timing regulation. To test whether the GFP-tagged H3.1 is incorporated into nucleosomes upon targeting to the PCH, we performed fluorescence recovery after photobleaching (FRAP) experiments in three conditions: GFP-tagged MaSat-binding protein (MajSat-GFP) (Lindhout et al, 2007) as a negative control, along with targeted GFP-H3.1 K36, or GFP-H3.1 K36M (Fig. EV4B). Upon photobleaching of the selected PCH, MajSat-GFP showed rapid fluorescence recovery, consistent with dynamic exchange and absence of chromatin incorporation. In contrast, both GFP-H3.1 K36 and K36M showed negligible recovery, indicating stable chromatin incorporation at PCH (Fig. EV4C,D; Movie EV1). These findings support the interpretation that the GFP-H3.1 fusion proteins are deposited into nucleosomes within PCH, enabling specific local modulation of H3K36 methylation. We further validated the approach by measuring the relative H3K9me3 and H3K36me3 levels in PCH for K36 and K36M. We used 3D images immunostained for H3K9me3 and H3K36me3 in cells with relatively low DNA content (G1/ early S) and the enrichment of GFP at the PCH. We observed that, while the level of H3K9me3 was slightly reduced but comparable, the level of H3K36me3 was significantly reduced in the H3.1 K36M-targeted cells (Fig. EV4E,F). These results validate the targeted approach to reduce the H3K36me3 level in PCH. We further dissected the replication progression using the pulse-chase method as earlier (Fig. EV3E). In the H3.1 K36 targeted cells, we observed the euchromatin replication, followed by the replication of PCH in S II and nuclear periphery nucleolar-associated chromatin, which were replicated in S III. However, upon targeting the H3.1 K36M, we observed a nuanced replication progression as observed earlier upon the knockdown of KMTs. While the euchromatin was replicated early in the S I, PCH replication was changed, leading to concomitant replication of constitutive and facultative heterochromatin as inferred from the RFi localization associated with both PCH and nuclear periphery/nucleolar associated chromatin domains at a given time (Fig. 2A). To investigate how H3.1 K36M targeting affects replication timing at pericentromeric heterochromatin (PCH), we first used the 'nucim' pipeline to map replication foci (RFi) to chromatin compaction classes in S II. In H3.1 K36M-targeted cells, we observed a relative shift of replicating chromatin from the INC towards ANC, indicating altered spatial replication patterns (Fig. 2B). To further validate this, we measured mean EdU intensity within PCH across S-phase stages. In control cells expressing H3.1 K36, EdU intensity peaked sharply in S II, reflecting the expected timing of PCH replication (Fig. 2C). In contrast, cells expressing H3.1 K36M showed a broader and relatively lower enrichment of EdU across both S II and S III, suggesting a loss of tight replication timing and a possible temporal spreading of PCH replication. These findings were further

validated by live-cell time-lapse imaging, which revealed that, upon H3.1 K36M incorporation, the PCH replication duration increased approximately 1.6-fold as compared to control cells. In H3.1 K36M-expressing cells, PCH replication extended across both S II and S III stages and overlapped with the replication of nuclear and nucleolar lamina-associated chromatin. Additionally, replication of perinuclear and perinucleolar regions appeared to initiate approximately 1.6 h earlier in H3.1 K36M cells compared to controls. As a result, the combined S II and S III replication period increased to an average of 4.04 h. (Fig. 2D,E; Movies EV2 and EV3). These observations suggest that disruption of H3K36me3 dynamics not only alters the timing and coordination of heterochromatin replication but also erases the temporal separation between the replication of these two distinct heterochromatin domains.

These observations suggest that H3K36me3 dynamically associates with PCH before replication, modulating its replication timing. Reducing H3K36me3 globally and locally at PCH leads to aberrations in the replication timing of heterochromatin. These observations suggest that the dynamic nature of H3K36me3 is crucial in regulating chromatin duplication, as altered replication timing often disrupts chromatin reestablishment and affects genome stability (Heinz et al, 2018; Klein et al, 2021).

## H3K36me3 depletion reduces MajSat transcripts and alters transcriptional activity in pericentromeric heterochromatin

While we observed the dynamic H3K36me3 fine-tuning of the replication timing of PCH, the underlying mechanism was unclear. While its potential role in replication timing control was ambiguous, the role of H3K36me3 in transcription elongation is well studied. The tandem repeats of MajSat comprising the PCH are not generally highly expressed, but these regions are known to be transcribed in a developmentally/cell cycle-regulated manner. (Lu and Gilbert, 2007; Novo et al, 2022). The MajSat RNA plays a crucial role in establishing and maintaining constitutive heterochromatin. These ncRNAs are required to form the PCH at the two-cell stage, and depletion of the strand-specific transcripts precludes the formation of chromocenters and development (Probst et al, 2010). Furthermore, it encourages the recruitment and stabilization of histone methyltransferases through the formation of DNA:RNA hybrids and promotes the formation of spatial compartments by recruiting and stabilizing histone methylases and heterochromatin proteins (Velazquez Camacho et al, 2017; Quinodoz et al, 2021). Hence, to elucidate the role of H3K36me3 in maintaining chromatin organization and in replication program regulation, we aimed to characterize the transcriptional activity and MajSat RNA in the pericentromeric heterochromatin.

To investigate whether MajSat RNA expression fluctuates during the cell cycle, we quantified forward and reverse MaSat transcripts across G1, S-phase substages, and G2 using RNA FISH with fluorescently labeled strand-specific locked nucleic acid (LNA) probes (Probst et al, 2010). Forward strand expression peaked during S I, coinciding with the H3K36me3 peak (Fig. 1D), along with S III, and was significantly reduced in G2. Reverse strand expression was more stable but also showed a modest accumulation as the cell cycle progressed and peaked in G2 (Fig. EV5A,B). These data suggest a dynamic, strand-specific regulation of MajSat transcription during cell cycle progression, with forward transcripts showing higher expression before and after PCH replication. Then, we measured the effect of loss of H3K36me3 on MajSat RNA

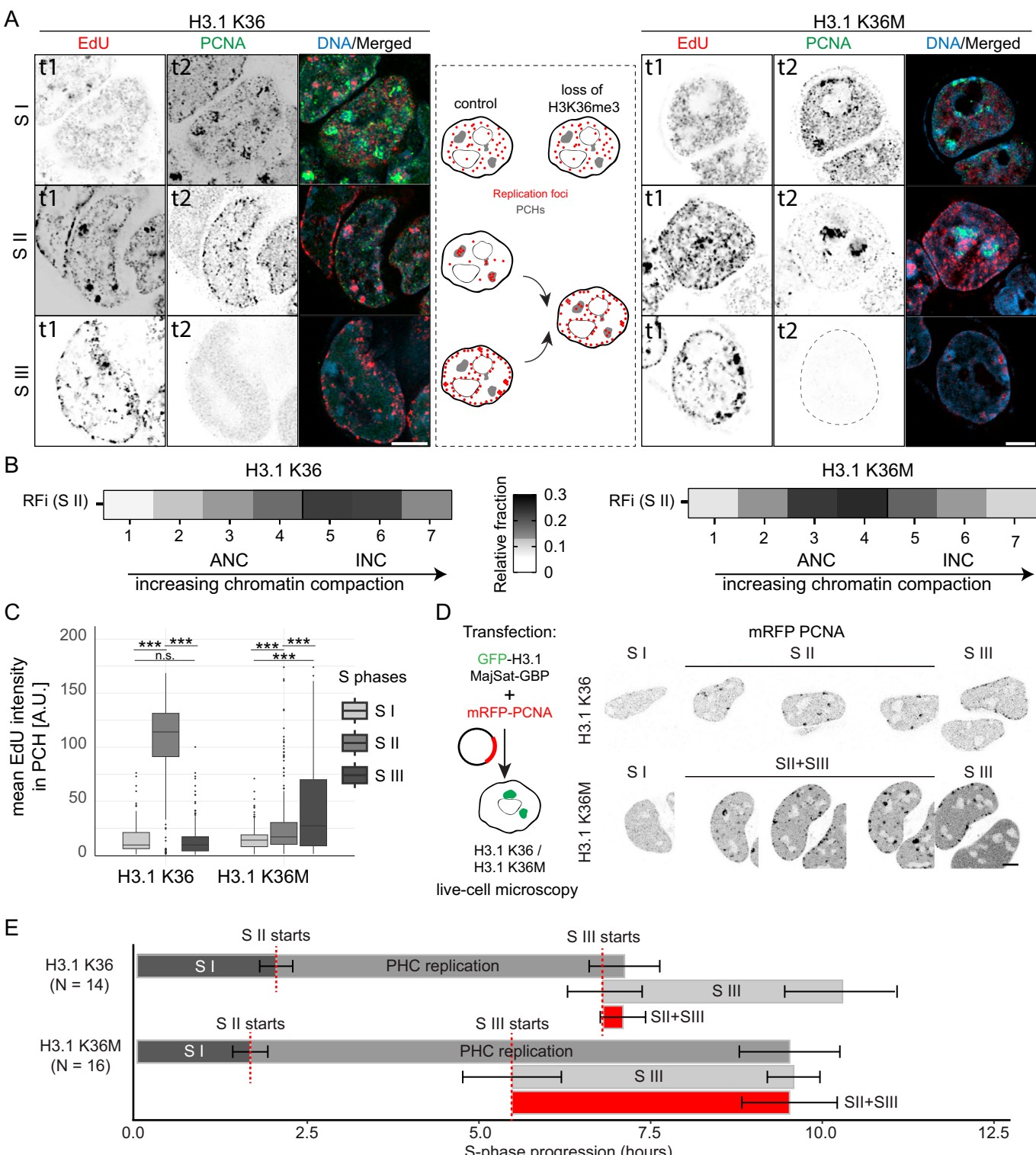

expressions separately using RNA FISH and qPCR after reverse transcription of total RNA (RT-qPCR) in cells transfected with esiRNA against eGFP (negative control) and KMTs (SETD2 + NSD1) (Fig. 3A,B). We observed a significant drop in the MajSat RNA level upon the knockdown of KMTs as inferred from both RNA FISH and qPCR. We further quantified the relative level of

MajSat RNA, where the level of H3K36me3 was locally reduced from PCH by targeting the GFP-tagged H3.1 K36M through the GBP-MajSat PZF. Targeting H3.1 K36M to the PCH resulted in a relative drop in MajSat RNA compared to H3.1 K36 targeting (Fig. 3C). We went on to characterize the transcriptional activities in the PCH under these disrupted conditions by directly analyzing the

**Figure 2. Disrupting the H3K36me3 dynamics in the pericentromeric heterochromatin (PCH) leads to replication timing aberration.**

(A) The images show the order of RFi patterns inferred from the pulse-chase experiment. In H3.1 K36, the replication progresses from euchromatin to constitutive heterochromatin, followed by lamin-associated domains. In H3.1 K36M, pericentromeric and lamin-associated chromatin domains are replicated concomitantly after euchromatin is replicated. The illustration in the middle represents the change in the RFi pattern with the loss of H3K36me3 globally and locally from PCH. (B) The heatmap shows the fraction of replication foci (RFi) using EdU mapped to the compaction classes in S II (for the K36M, the S II cells were selected visually, where a large fraction of PCH is replicating, and S III were selected where the large fraction of the nucleolar/nuclear lamin-associated chromatin were replicating and with no punctuated foci from PCNA channel). ANC active nuclear compartment, INC inactive nuclear compartment. (C) The mean EdU intensity in the PCH in S phase stages was measured by segmenting the targeted H3.1 GFP to PCH for both H3.1 K36 & K36M. The statistical test was performed using ANOVA followed by Tukey's honest significant difference (HSD). The p-values for H3.1 K36 are: S I: S II $= 0.00000018$, S I: S III $= 0.9907532$, S II: S III $= 0.00000034$, and for H3.1 K36M are: S I: S II $= 0.0000001$, S I: S III $= 0.0000000016$, and S II: S III $= 0.000001$. The boxplot shows the median (line), interquartile range (box: 25th to 75th percentile), and whiskers extending to data within $1.5\times$ the IQR. n.s. not significant; $*p < 0.05$; $**p < 0.005$; $***p < 0.0005$. $n$ (H3.1 K36) $= 26$, $n$ (H3.1 K36M) $= 31$ (both from three biological replicates). (D) Live cells with PCH-targeted GFP-H3.1 and expressing mRFP-PCNA were imaged at 20 min intervals for several hours. Representative videos are shown in Movies EV2 and EV3. Scale bar: 5 μm. (E) Barplots show the average time for each S-phase substage in hours. The whiskers represent the standard error with a 95% confidence interval. The observed time for S II $+$ S III concomitant patterns is shown in red. $n$ (K36) $= 15$, $n$ (K36M) $= 17$ (both from three biological replicates). Source data are available online for this figure.

transcriptional machinery. We measure the active transcription by quantifying the Ser5 phosphorylation of the C-terminal domain of RNA polymerase II (Ser5 RNA Pol II). After immunostaining, we performed super-resolution imaging and 3D image analysis to quantify active transcription sites in the PCH. After segmenting the nuclei, the subnuclear DAPI-dense PCH were segmented using a threshold- and size-selection-based approach. The quality of the segmented regions was verified visually. In parallel, 3D spots of Ser5 RNA Pol II were also segmented, and the number of total spots in PCH was counted in individual cells in different conditions. We observed that the number of spots of Ser5 RNA Pol II increased upon knockdown of KMTs compared to the negative control (Fig. 3D). After locally reducing the levels of H3K36me3 in PCH, we observed a similar trend where targeting H3.1 K36M to the PCH increased the number of Ser5 RNA Pol II relative to control (Fig. 3E). While the Ser5 modification of RNA pol II is present at the promoter regions marking active transcription, the phosphorylation of Ser2 marks the transcription elongation (Jeronimo et al, 2013). To further investigate transcriptional changes at PCH upon loss of H3K36me3, we quantified both Ser5 and Ser2-phosphorylated RNA Pol II in the same cells (Fig. 3F). Targeting H3.1 K36M to PCH led to a notable increase in Ser5 signal intensity, while Ser2 signal was relatively reduced. This uncoupling of initiation and elongation markers suggests a transcriptional imbalance, potentially indicative of polymerase stalling or inefficient transition into elongation at pericentromeric heterochromatin.

Our findings suggest that H3K36me3 promotes transcription fidelity in pericentromeric heterochromatin, preventing premature transcription initiation while promoting elongation. Its loss disrupts this balance, increasing the recruitment of transcription initiation machinery but reducing productive MajSat RNA synthesis, possibly impacting chromatin organization and replication timing.

## MajSat forward RNA impacts replication timing of constitutive heterochromatin and genome stability

Given its established role in transcription regulation and its impact on replication timing in PCH, we hypothesized that H3K36me3-mediated expression of MajSat RNA transcripts might contribute to the regulation of replication timing. Although the relative expression level of MajSat RNA is significantly lower in mESCs compared to the one- or two-cell embryonic stages, transcription is not entirely abolished, suggesting a functional role beyond early development (Probst et al,

2010; Huo et al, 2020). To assess this, we selectively disrupted MajSat RNA function using locked nucleic acid (LNA) DNA gapmers, a method that not only induces RNA degradation but also impairs RNA folding and disrupts RNA-protein or RNA-DNA interactions (Mayer et al, 2006). Given the distinct strand-specific roles of MajSat transcripts in PCH organization and maintenance, we used gapmers targeting either the forward or reverse MajSat transcript, while an LNA gapmer against eGFP served as a negative control. (Probst et al, 2010; Casanova et al, 2013). We first validated the knockdown using quantitative PCR (Fig. EV5C), with forward and reverse MajSat being 17 and 25% downregulated, which was comparable to their abundance changes during replication (12 and 17%, respectively) (Fig. EV5A,B). To assess whether loss of MajSat RNA affects chromatin, we next measured levels of H3K9me3 and H3K36me3 at pericentromeric heterochromatin (PCH) following knockdown. H3K9me3 was significantly reduced in both forward and reverse knockdown conditions, with a more pronounced effect in the reverse knockdown (Fig. EV5D). Similarly, H3K36me3 levels were also reduced, particularly in the reverse knockdown, whereas in the forward knockdown, the reduction was less pronounced (Fig. EV5E). These findings support that MajSat transcripts contribute to the maintenance of repressive chromatin at PCH, specifically the reverse MajSat RNA. Following the disruption, we analyzed the replication program using a pulse-chase approach with a 3-hour chase duration post-EdU labeling (Fig. EV3E). Immunostaining for EdU and PCNA allowed us to infer the spatiotemporal replication dynamics of different chromatin domains. In cells treated with the eGFP (control) or the reverse MajSat RNA gapmer, replication followed the expected sequential program: euchromatin replicated during S I, followed by PCH in S II, and nuclear periphery/nucleolar-associated chromatin in S III (Fig. 4A,B). However, targeting the forward MajSat RNA led to a delayed replication timing of the PCH. Instead of the typical sequential pattern, we observed a concomitant replication of PCH and nuclear periphery/nucleolar-associated chromatin after the euchromatin was replicated in S I (Fig. 4C). Mapping the RFi to chromatin compaction classes in S II revealed a shift in the relative distribution of replicating chromatin toward the active nuclear compartment (ANC) in forward MajSat knockdown cells. This shift likely reflects the concurrent replication of additional, less compact chromatin types alongside PCH. To quantify this shift, we measured EdU incorporation specifically within PCH across S-phase substages. In control and reverse MajSat knockdown cells, EdU signal peaked in S II, consistent with the confined timing of PCH replication (Fig. 4D). In contrast, forward MajSat RNA knockdown significantly reduced EdU intensity in S II and increased

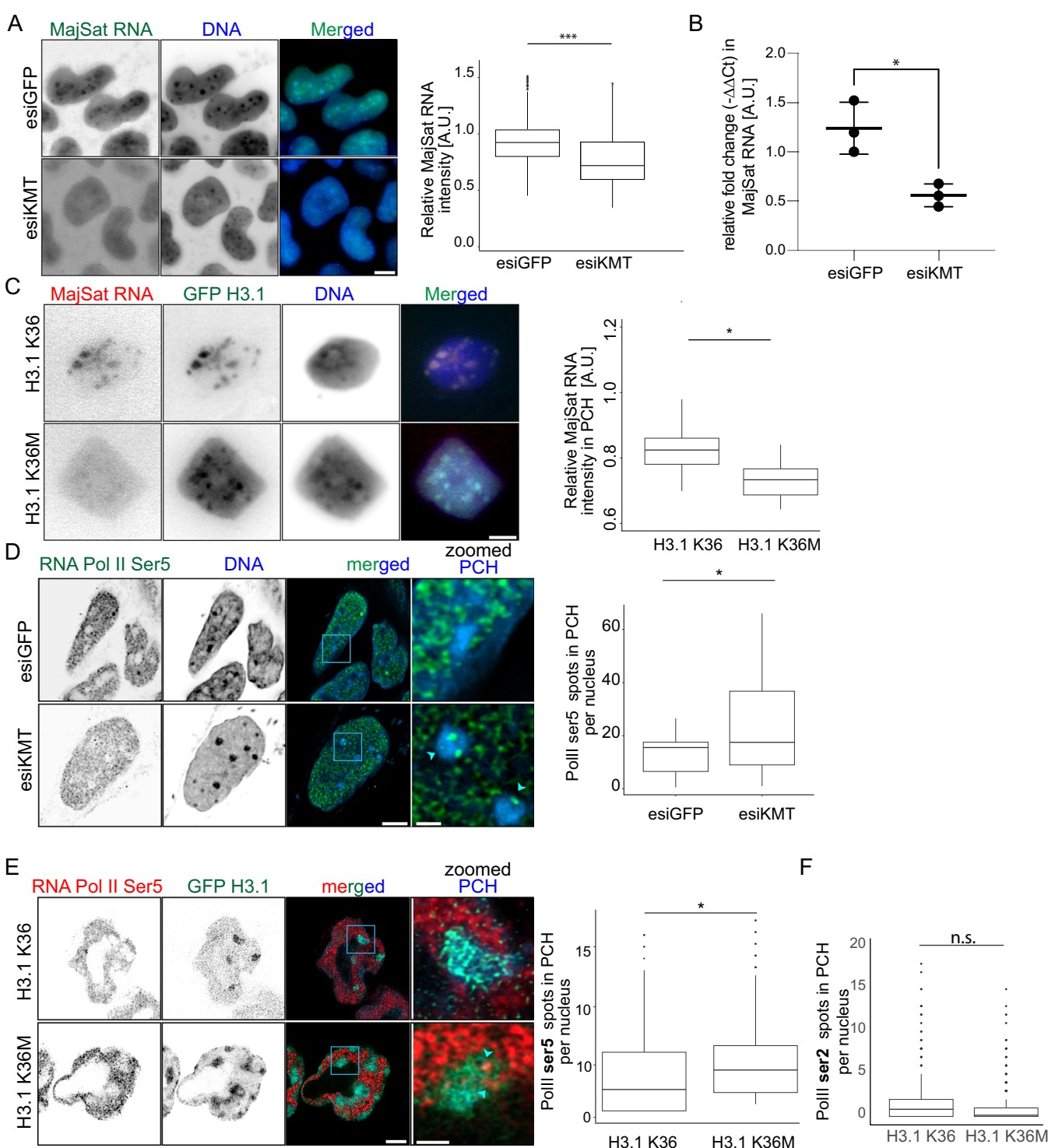

it in S III, indicating that PCH replication was delayed and extended into later stages. These results suggest that forward MajSat RNA is necessary to maintain the precise replication timing of PCH, and that its loss disrupts the spatiotemporal coordination of constitutive heterochromatin replication, leading to overlap with the replication timing of facultative heterochromatin domains. Although the measured depletion of forward MajSat RNA upon LNA-gapmer interference was modest (<20%), likely reflecting technical limitations

in targeting and quantifying highly repetitive transcripts, this partial reduction was consistently associated with significant replication timing defects at PCH (Fig. 4C,D). This suggests that replication timing is highly sensitive to forward MajSat RNA levels, and even modest reductions can impair proper progression.

This aberration closely mirrored the replication defects seen upon the global or local depletion of H3K36me3 at PCH, suggesting that forward MajSat RNA plays a crucial role in maintaining the

**Figure 3.   H3K36me3 influences the MajSat expression and transcriptional machinery functions.**

(A) Images and the boxplot show the loss of MajSat RNA as inferred from the normalized MajSat RNA FISH (both forward and reverse) signals inside the pericentromeric heterochromatin upon loss of H3K36me3. The p value inferred from the Wilcoxon test <2.2e-16. n (esiGFP) = 182, n (esiKMT) = 254. For both conditions, cells were imaged from three biological replicates. (B) The plot shows the level of MajSat RNA in esiRNA and esiKMT. The level of MajSat RNA is reduced with the loss of H3K36me3. Three independent biological replicates were performed, with the p-value inferred from the Wilcoxon test = 0.0309. (C) A similar loss of MajSat RNA was observed in the pericentromeric heterochromatin upon H3.1 K36M targeting. Images show the level of MajSat transcript in the cells where H3.1 is targeted to the PCH. The boxplot shows a reduction in the normalized level of MajSat transcript upon PCH targeted with H3.1 K36M. The p-value inferred from the Wilcoxon test = 0.038. n (H3.1 K36) = 15, n (H3.1 K36M) = 15 (from three biological replicates). (D) The plot shows the RNA pol II ser5 (associated with promoter of active transcription) spots increase in the pericentromeric heterochromatin upon the loss of H3K36me3, suggesting increased transcription stalling. n (esiGFP) = 21, n (esiKMT) = 26. (cells were imaged from three biological replicates) The p value inferred from the Wilcoxon test = 0.042. (E) The phosphorylation of Ser5 (promoter) of RNA pol II CTD was detected, and the number of 3D spots inside the PCH was quantified using a segmented GFP dense region as a region of interest on Fiji. The p value inferred from the Wilcoxon test = 0.04444282. n (H3.1 K36) = 16, n (H3.1 K36M) = 21 (cells were imaged from three biological replicates). (F) The boxplot shows the changes in 3D spots inside the PCH of phosphorylation of Ser2 (elongation) of RNA pol II CTD. n (H3.1 K36) = 16, n (H3.1 K36M) = 21. The p value inferred from the Wilcoxon test = 0.2631529. (cells were imaged from three biological replicates). The boxplot shows the median (line), interquartile range (box: 25th to 75th percentile), and whiskers extending to data within 1.5× the IQR. n.s. not significant; *p < 0.05; **p < 0.005; ***p < 0.0005. Scale bar: 5 and 2 μm in the zoomed images. Source data are available online for this figure.

proper replication timing of these domains in mESCs. Given that H3K36me3 is known to promote transcription elongation, its effect on replication timing may be mediated through MajSat RNA expression. The dynamic localization of H3K36me3 at these regions may, therefore, suggest a sequential model where H3K36me3 loss reduces MajSat transcripts, but only forward-strand RNA contributes directly to replication timing. Whereas reverse strand RNA primarily affects epigenetic state by reducing both H3K9me3 and H3K36me3, without strongly impacting replication timing.

While histone modifications, particularly acetylation, are known to directly influence the replication timing, long non-coding RNAs are reported to regulate the replication timing on the individual chromosome level (Casas-Delucchi et al, 2012b, 2011; Donley et al, 2013, 2015). While the exact mechanism through which the replication timing is disrupted upon loss of MajSat forward transcript remains to be elucidated, its impact aligns with previous findings on strand-specific non-coding RNA regulation of replication timing (Platt et al, 2018).

Classic heterochromatin marks such as H3K9me3 and H4K20me3 have been correlated with PCH structure and replication timing. For instance, H4K20me2 is recognized by ORC1 to promote replication origin licensing, and H2A.Z has been shown to facilitate H4K20me2 deposition at origins (Kuo et al, 2012; Long et al, 2020). H4K20me3, deposited by Suv420h1/h2, ensures timely heterochromatin replication at late-firing origins, and its di-/tri-methylation state also influences MCM complex loading (Shoaib et al, 2018; Hayashi-Takanaka et al, 2021). Nonetheless, depleting these histone methylation marks did not directly affect replication timing, albeit regulating histone acetylation level changed replication timing of heterochromatin (Rausch et al, 2020) (Casas-Delucchi et al, 2012b). Our data suggest that H3K36me3 may act in parallel with these pathways and indirectly through MajSat RNA. Furthermore, H3K36 demethylases such as KDM2A, known to interact with HP1α, could contribute to the turnover of H3K36me3 at PCH (Blackledge et al, 2010; Borgel et al, 2017). Although we did not directly test the role of demethylases, the observed loss of H3K36me3 post-replication and its dynamic enrichment prior to replication suggest a tightly regulated deposition/removal cycle. This regulation may involve coordinated action between SETD2/ NSDs and demethylases like KDM2A, whose chromatin binding could be mediated by HP1 or satellite RNA. Comparing the two experimental strategies used to reduce H3K36me3, which are global knockdown of methyltransferases (esiKMT) and PCH-specific

targeting of H3.1 K36M, revealed that both approaches led to disrupted replication timing, and the effects were pronounced with the local, replication-coupled H3.1 K36M targeting. This suggests that precise spatial regulation of H3K36me3 at PCH is critical, and that its local loss is sufficient to trigger transcriptional, replicative, and genome stability changes.

Given the crucial role of PCH in maintaining genome stability, we investigated whether the H3K36me3 depletion contributes to global genome instability (Jagannathan et al, 2018). Depletion of SETD2 leads to a loss of H3K36me3, which has been identified as an early trigger of genome instability (Smirnov et al, 2024). Furthermore, the MajSat RNA expression itself varies and is known to modulate the heterochromatin condensates to safeguard chromosome stability in pluripotent stem cells (Tosolini et al, 2018; Novo et al, 2022; Quinodoz et al, 2021). To assess whether loss of H3K36me3 at PCH induces genome instability, we measured γH2AX levels, an established marker of DNA damage (Rogakou et al, 1999; Sedelnikova et al, 2002), in cells targeted with H3.1 K36 or H3.1 K36M. Targeting the dominant-negative K36M mutant resulted in a significant increase in γH2AX signal both in the whole nucleus and at PCH regions (Fig. EV5F,G), indicating that local disruption of H3K36me3 at chromocenters compromises genome integrity. Knocking down the MajSat transcripts using gapmers was shown to increase genome instability, structural changes in chromosomes, and defective cohesions (Novo et al, 2022). Hence, in parallel, we analyzed micronuclei formation following strand-specific knockdown of MajSat RNA. Micronuclei, which serve as a readout of chromosomal instability, were significantly more frequent upon forward strand depletion compared to control or reverse knockdown (Fig. EV5H). These findings suggest that both H3K36me3 and forward MajSat RNA contribute to maintaining genome stability, likely by ensuring proper replication timing and chromatin organization at PCH. Future work should aim to elucidate the cell cycle and chromatin-specific dynamics of micronuclei formation.

## Conclusion

Together, our findings support a model in which dynamic enrichment of H3K36me3 at pericentromeric heterochromatin safeguards genome stability by regulating replication timing and maintaining chromatin integrity. H3K36me3 promotes the expression of major satellite (MajSat) RNAs, and our results reveal a unique strand-specific role of the forward transcript in ensuring the timely replication of constitutive heterochromatin. Although the mechanistic details of how H3K36me3-

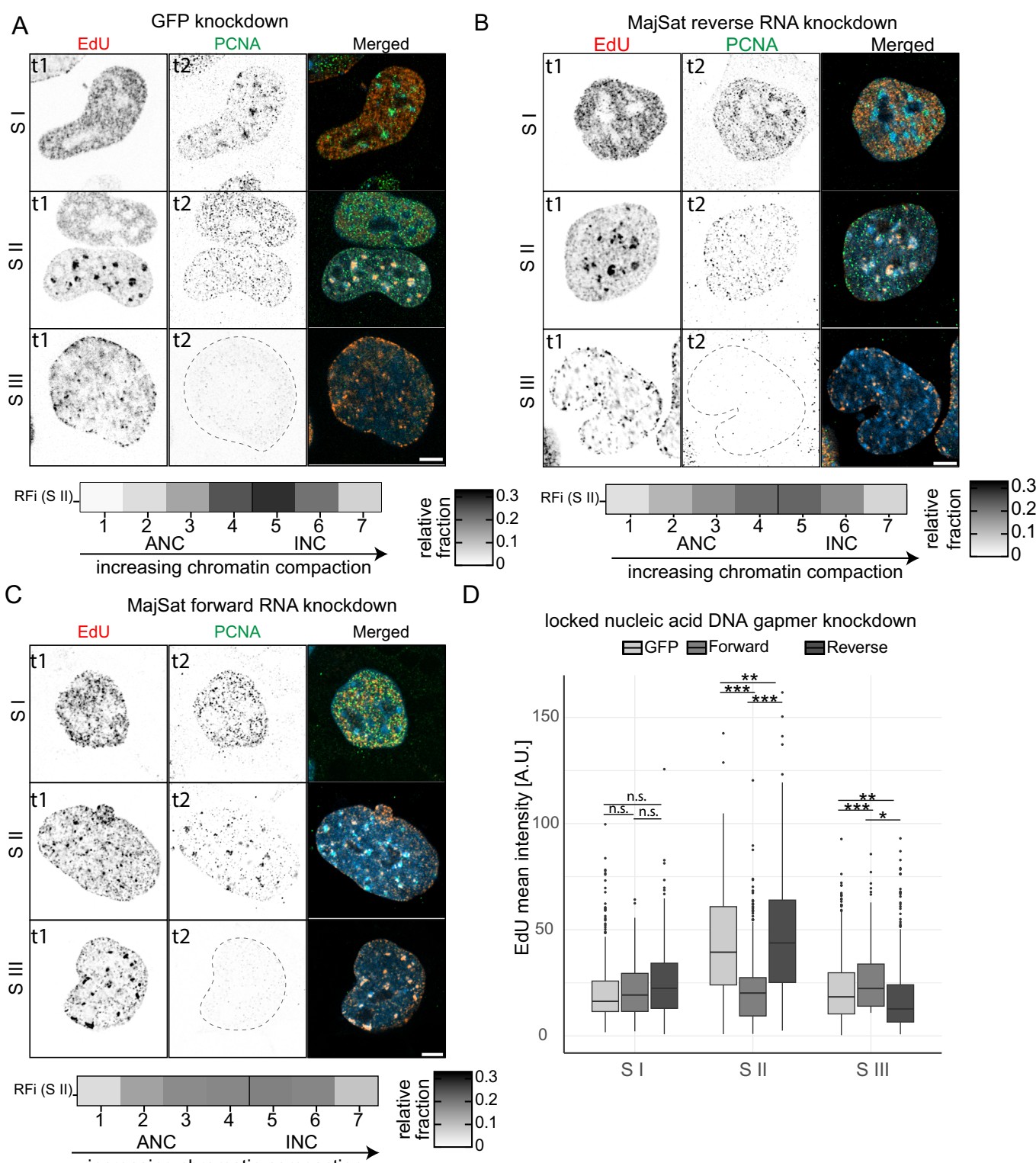

**A** GFP knockdown
EdU  PCNA  Merged

**B** MajSat reverse RNA knockdown
EdU  PCNA  Merged

**C** MajSat forward RNA knockdown
EdU  PCNA  Merged

**D** locked nucleic acid DNA gapmer knockdown

driven transcription of MajSat RNA enforces replication timing remain unresolved, our data indicate a sequential relationship in which H3K36me3 loss reduces MajSat RNA levels, while only the forward strand directly supports proper replication timing. In contrast, reverse strand depletion preferentially impacts chromatin marks (H3K36me3

and H3K9me3) without strongly altering replication timing, suggesting distinct strand-specific functions. Importantly, even modest reductions of forward MajSat RNA are sufficient to impair replication timing, underscoring the sensitivity of these domains to non-coding RNA regulation. Thus, our work highlights an epigenetic pathway where

**Figure 4. Forward MajSat RNA regulates the replication timing of the pericentromeric heterochromatin.**

The locked nucleic acid (LNA) DNA gapmers were used to achieve targeted degradation of specific RNA, followed by a pulse (EdU, t1, red) and a chase, followed by detection of EdU and PCNA (t2, green), capturing two sub-S phase RFi patterns in the same cell. The heatplots (in **A–C**) show the RFi mapped to compaction classes in S II. (**A**) The LNA-DNA gapmers against eGFP were used as a negative control. Scale bar: 5 µm. (**B**) The interference of MajSat reverse RNA did not affect the typical progression of replication. Scale bar: 5 µm. (**C**) The interference of MajSat forward RNA leads to the delayed replication of the PCH, resulting in a concomitant replication of constitutive and facultative heterochromatin. Scale bar: 5 µm. (**D**) The plot shows the mean EdU intensity inside pericentromeric heterochromatin (PCH) in S-phase stages. The statistical test among LNA conditions in each S phase stage was performed using ANOVA followed by Tukey's honest significant difference (HSD) and extracted. The p-values are: S II (GFP:forward = 0.000000011, GFP:reverse=0.0027602, forward:reverse = 0.000000345), S III (GFP:forward = 0.00000785, GFP:reverse = 0.0010184, forward:reverse = 0.046895). n (GFP) = , n (forward) = , n (reverse) =. Three biological replicates. The boxplot shows the median (line), interquartile range (box: 25th to 75th percentile), and whiskers extending to data within 1.5× the IQR. n.s. not significant; *$p < 0.05$; **$p < 0.005$; ***$p < 0.0005$. Source data are available online for this figure.

H3K36me3 influences replication timing through its effect on MajSat transcription, while also outlining open questions about the molecular mechanisms that couple satellite RNA with replication dynamics.

# Methods

## Reagents and tools table

| Experimental models: cell lines | | |
| --- | --- | --- |
| mESC J1 | Li et al, 1992 | |
| **Recombinant DNA** | **Reference or source** | **Identifier or Catalog Number** |
| pEGFP-C1 | Clontech Laboratories | 6084-1 |
| EGFP-H3.1 | This study | pc2099, Addgene ID: 241429 |
| EGFP-H3.1 K36M | This study | pc5139, Addgene ID: 241430 |
| pGBP MajSat | Casas-Delucchi et al. 2012a | pc2469 |
| **Antibodies** | **Host/Clonality/Dilution** | **Catalog Number & Source** |
| Anti-PCNA | Mouse Monoclonal,1:200 | ab29, Abcam, Cambridge, UK |
| Anti-H3 | Rat Monoclonal, 1:1000 (WB*) | 6164, Active Motif, Waterloo, Belgium |
| Anti-H3K9me3 | Mouse Monoclonal, 1:200 | 39285, Active Motif, Waterloo, Belgium |
| Anti-H3K36me3 | Rabbit, Polyclonal, 1:2000 | ab9050, Abcam, Cambridge, UK |
| Anti-H3K27me3 | Mouse, Monoclonal,1:200 | 61017, Thermo Fisher Scientific, Waltham, MA, USA |
| Anti-H3K9ac | Rabbit Polyclonal, 1:200 | 39917, Active Motif, Waterloo, Belgium |
| Anti-H3K4me3 | Rabbit Polyclonal, 1:200 | 39159, Active Motif, Waterloo, Belgium |
| Anti-H4K20me3 | Rabbit Polyclonal, 1:500 | ab9053, Abcam, Cambridge, UK |
| Anti-H3K9me2 | Mouse Monoclonal, 1:500 | 39683, Active Motif, Waterloo, Belgium |
| Anti-H4K5ac | Rabbit Polyclonal, 1:500 | ab51997, Abcam, Cambridge, UK |
| Anti-H4K8ac | Rabbit Polyclonal, 1:200 | ab15823, Abcam, Cambridge, UK |
| Anti-H4K12ac | Mouse Monoclonal, 1:200 | 61527, Active Motif, Waterloo, Belgium |
| Anti-H4K16ac | Rabbit Polyclonal, 1:200 | 39168, Active Motif, Waterloo, Belgium |
| Anti-H3K56ac | Rabbit Monoclonal, 1:250 | 2134-1, Epitomics Inc., Burlingame, California (Now Abcam) |
| Anti-NSD1 Antibody, clone 1NW-1A10 | Mouse Ascites, 1:250 | 04-1565, Sigma-Aldrich Chemie GmbH, Steinheim, Germany |
| Anti-SETD2 | Rabbit Monoclonal, 1:250 | E4W8Q, Cell Signaling Technology, Inc., MA, USA |
| Anti-γH2AX | Mouse Monoclonal, 1:800 | 3601680, Merck, Darmstadt, Germany |
| Anti-RNA polymerase II RPB1 phospho S5 | Mouse Monoclonal, 1:200 | ab5408, Abcam, Cambridge, UK |
| Anti-RNA polymerase II RPB1 phospho S2 | Rat Monoclonal, 1:500 | 61084 Active Motif, Waterloo, Belgium |
| Anti-mouse IgG Alexa Fluor 488 | Goat Polyclonal, 1:500 | A11029, Thermo Fisher Scientific, Waltham, MA, USA |
| Anti-rat IgG Cy3 | Donkey Polyclonal, 1:400 | 163464, Jackson ImmunoResearch Europe Ltd., Cambridge, UK |
| Anti-rat IgG Cy5 | Donkey Polyclonal, 1:1000 (WB*) | A10525, Thermo Fisher Scientific, Waltham, MA, USA |
| Anti-mouse IgG Cy5 | Donkey Polyclonal, 1:200 | JIM-715-175-150, Jackson ImmunoResearch Europe Ltd., Cambridge, UK |
| Anti-rabbit IgG Alexa Fluor 488 | Goat Polyclonal, 1:500 | A-11034, Thermo Fisher Scientific, Waltham, MA, USA |
| Anti-rabbit IgG Cy3 | Donkey Polyclonal, 1:1000 (WB*) | JIM-711-165-152 Jackson ImmunoResearch Europe Ltd., Cambridge, UK |
| **Primers/LNA Gapmers** | **Sequence (5'-3')** | **Source** |
| H3.1 Forward | AAGAATTCAATGGCTCGTACGAAGCAAAC | Sigma-Aldrich Oligo, Darmstadt, Germany |
| H3.1 Reverse | AAGCGGCCGCTGCCCTTTCCCCACGGATG | Sigma-Aldrich Oligo, Darmstadt, Germany |
| H3.1 K36M Forward | ACCGGTGGCGTG**ATG**AAACCTCATCGC | Sigma-Aldrich Oligo, Darmstadt, Germany |
| H3.1 K36M Reverse | GCGATGAGGTTT**CAT**CACGCCACCGGT | Sigma-Aldrich Oligo, Darmstadt, Germany |
| LNA DNA gapmer GFP | gagaAAGTGTGACAagtg | Sigma-Aldrich Oligo, Darmstadt, Germany |
| LNA DNA gapmer maj 1 (Forward) | acatCCACTTGACGActtg | Sigma-Aldrich Oligo, Darmstadt, Germany |
| LNA DNA gapmer maj 2 (Reverse) | tattTCACGTCCTAAagtg | Sigma-Aldrich Oligo, Darmstadt, Germany |
| qPCR MajSat Forward | GACGACTTGAAAAATGACGAAATC | Sigma-Aldrich Oligo, Darmstadt, Germany |
| qPCR MajSat Reverse | CATATTCCAGGTCCTTCAGTGTGC | Sigma-Aldrich Oligo, Darmstadt, Germany |
| qPCR GAPDH Forward | ATCACTGCCACCCAGAAGACTGTGGA | Sigma-Aldrich Oligo, Darmstadt, Germany |
| qPCR GAPDH Forward | GAGCTTGACAAAGTTGTCATTGAGAGC | Sigma-Aldrich Oligo, Darmstadt, Germany |
| FISH_MajSat Forward (major 1) | TCTTGCCATATTCCACGTCC | Sigma-Aldrich Oligo, Darmstadt, Germany |

| FISH_MajSat Reverse (major 2) | GCGAGGAAAACTGAAAAAGG | Sigma-Aldrich Oligo, Darmstadt, Germany |
|---|---|---|
| FISH_MajSat Forward (major 3) | GATTTCGTCATTTTTCAAGT | Sigma-Aldrich Oligo, Darmstadt, Germany |
| FISH_MajSat Reverse (major 4) | GCGAGAAAACTGAAAATCAC | Sigma-Aldrich Oligo, Darmstadt, Germany |
| **Chemicals, enzymes, and other reagents** | **Reference or source** | **Identifier or Catalog Number** |
| ALLin™ HiFi DNA Polymerase | highQu GmbH, Germany | HLE0201 |
| Kinase (T4 Polynucleotide Kinase (PNK)) | New England Biolabs GmbH, Frankfurt am Main, Germany | M0201L |
| Ligase (T4 DNA Ligase) | New England Biolabs GmbH, Frankfurt am Main, Germany | M0202L |
| DpnI | New England Biolabs GmbH, Frankfurt am Main, Germany | R0176L |
| DMEM high glucose | Sigma-Aldrich Chemie GmbH, Steinheim, Germany | D6429 |
| 1× non-essential amino acids | Sigma-Aldrich Chemie GmbH, Steinheim, Germany | M7145 |
| 1×penicillin/ streptomycin | Sigma-Aldrich Chemie GmbH, Steinheim, Germany | P4333 |
| 1× ʟ-glutamine | Sigma-Aldrich Chemie GmbH, Steinheim, Germany | G7513 |
| 0.1 mM beta-mercaptoethanol | Carl Roth, Karlsruhe, Germany | 4227 |
| 1000 U/ml recombinant mouse LIF (Millipore) | Merck KGaA, Darmstadt, Germany | ESG1107 |
| 2i (1 M PD032591 and 3 M CHIR99021) | Axon Medchem, Netherlands | 1408 and 1386 respectively |
| Gelatin | Sigma-Aldrich Chemie GmbH, Steinheim, Germany | G1393 |
| Laminin | Geyer GmbH & Co. KG, Renningen, Germany | L2020-1MG |
| Lipofectamine 3000 | Thermo Fisher Scientific, MA, USA | L3000015 |
| Phosphate-buffered saline | Sigma-Aldrich, Merck KGaA, Darmstadt, Germany | P3813 |
| Tris-Cl | Sigma-Aldrich, Merck KGaA, Darmstadt, Germany | T6664 |
| KCl | Sigma-Aldrich, Merck KGaA, Darmstadt, Germany | P9541 |
| MgCl₂ | Sigma-Aldrich, Merck KGaA, Darmstadt, Germany | M8266 |
| DTT | Sigma-Aldrich, Merck KGaA, Darmstadt, Germany | D0632 |
| $H_2SO_4$ | Carl Roth GmbH & Co. KG, Karlsruhe, Germany | 4625.1 |
| Trichloroacetic acid (TCA) | Carl Roth GmbH & Co. KG, Karlsruhe, Germany | 4855.1 |
| Formaldehyde solution, 37% (w/v) | Carl Roth GmbH & Co. KG, Karlsruhe, Germany | 4979.1 |
| Pierce™ 660 nm Protein Assay Reagent | Thermo Fisher Scientific, Waltham, MA, USA | 22660 |
| Triton™ X-100 | Carl Roth GmbH & Co. KG, Karlsruhe, Germany | 3051.2 |
| Tween®-20 | Carl Roth GmbH & Co. KG, Karlsruhe, Germany | 9127.1 |
| BSA (bovine serum albumin), powder | Sigma-Aldrich, Merck KGaA, Darmstadt, Germany | A2153 |
| EdU Click-IT cell proliferation kit | Carl Roth GmbH & Co. KG, Karlsruhe, Germany | 7845.1 |
| 5-TAMRA-Azide | Jena Bioscience, Jena, Germany | CLK-FA008-1 |
| 4',6-diamidino-2-phenylindole (DAPI) | Sigma-Aldrich Chemie GmbH, Steinheim, Germany | D9542 |
| Vectashield | Vector Laboratories Inc., Burlingame, CA, USA | VEC-H-1000 |
| RNeasy Mini kit | QIAGEN GmbH, Hilden, Germany | 74104 |
| SuperScript™ IV First-Strand Synthesis | Invitrogen, U.S. | 18091050 |

| SYBR premix (Platinum™ SYBR™ Green qPCR SuperMix) | Thermo Fisher Scientific, MA, USA | 11744100 |
|---|---|---|
| Vanadyl Ribonucleoside Complex (VRC) | New England Biolabs GmbH, Frankfurt am Main, Germany | S1402S |
| Formamide | Sigma-Aldrich Chemie GmbH, Steinheim, Germany | D4551 |
| **Software** | **Source/Reference** | |
| FiJi | https://fiji.sc/ (Schindelin et al, 2012) | |
| StarDist (FiJi) | https://github.com/stardist/stardist-imagej (Schmidt et al, 2018) | |
| 3D suite (FiJi) | https://mcib3d.frama.io/3d-suite-imagej/ (Ollion et al, 2013) | |
| R | https://www.r-project.org/ | |
| Nucim (R) | https://bioimaginggroup.github.io/nucim/ (Schmid et al, 2017) | |
| Zen | https://www.zeiss.com/ | |
| Amersham Imager 600 | | |
| **Microscopes** | | |
| Nikon CREST mounted on TiE2 platform | Objectives: 20x (SPlan Fluor LWD DIC, air, NA: 0.7) or a 40X (Plan Apo λ DIC, air, NA: 0.95) | Nikon Instruments Inc., Japan |
| LSM 980 microscope with Airyscan 2 | Objectives:63x (C plan-apochromatic, NA: 1.4) or 40x (plan-apochromatic, NA: 0.95) | Carl Zeiss AG, Oberkochen, Germany |

*WB = dilution used for Western blot, the rest refers to the dilution used in immunofluorescence staining

## Cell culture and knockdown

The mESC J1 cells were grown in Dulbecco's modified Eagle's medium (DMEM) high glucose (Cat. No.: D6429, Sigma-Aldrich Chemie GmbH, Steinheim, Germany) supplemented with 15% fetal calf serum (FCS), 1× non-essential amino acids (Cat. No.:M7145, Sigma-Aldrich Chemie GmbH, Steinheim, Germany), 1×penicillin/ streptomycin (Pen/Strep) (Cat. No.:P4333, Sigma-Aldrich Chemie GmbH, Steinheim, Germany), 1× ʟ-glutamine (Cat. No.: G7513, Sigma-Aldrich Chemie GmbH, Steinheim, Germany), 0.1 mM beta-mercaptoethanol (Cat. No.: 4227, Carl Roth, Karlsruhe, Germany), 1000 U/ml recombinant mouse LIF (Millipore) and 2i (1 M PD032591 and 3 M CHIR99021 (Cat. Nos.: 1408 and 1386, respectively, Axon Medchem, Netherlands)) on gelatin-coated culture dishes (0.2% gelatin; Cat. No.: G1393, Sigma-Aldrich Chemie GmbH, Steinheim, Germany) or laminin-coated coverslips (10 μg/ml laminin; Cat. No.: L2020-1MG, Th. Geyer GmbH & Co. KG, Renningen, Germany) (Li et al, 1992).

Around $10^5$ mESC J1 cells were seeded for 24 hr in each well of the six-well plates. Amount of 500 nM of each esiRNA targeting mouse SETD2 (esiRNA_SETD2, MISSION® esiRNA, EMU214881, Merck KGaA, Darmstadt, Germany) and NSD1 (esiRNA_Nsd1, MISSION® esiRNA, EMU186591, Merck, Darmstadt, Germany) using Lipofectamine 3000 (L3000015, Thermo Fisher Scientific, MA, USA) according to the protocol provided by the manufacturer. Cells were incubated 24–72 h post-transfection, and knockdown efficiency was monitored at 24, 36, 48, and 72 h. An equal amount (500 nM) of esiRNA_GFP (esiRNA_GFP, MISSION® esiRNA, EHUEGFP, Merck KGaA, Darmstadt, Germany) was used as a negative control. The knockdown was validated using the immunofluorescence staining against SETD2 and NSD1 and

high-throughput image analysis. The effect of knockdown on H3K36me3 was validated on a western blot using acid-extracted histones according to the protocol (Shechter et al, 2007). The loss of H3K36me3 in the pericentromeric heterochromatin (PCH) was measured by measuring the H3K36me3 signal normalized to the DAPI signals (described in the image analysis section in detail).

To knock down the expression of MajSat RNA, strand-specific LNA-DNA gapmers (250 nM) were transfected with Lipofectamine 3000 to cells seeded for 24 h according to the protocol provided by the manufacturer. After 24 h, the cells were washed, transfected again, and incubated for 12 h. Cells were washed and incubated in fresh media for 2 h before being used for further experiments. The gapmers used are LNA DNA gapmer maj 1 (Forward), LNA DNA gapmer maj 2 (Reverse), and LNA DNA gapmer gfp—as a control (Reagents and Tools Table) (Probst et al, 2010).

## Molecular cloning and targeting strategy

An expression vector encoding the human histone H3.1 was amplified from human cDNA using primers H3.1 Forward and H3.1 Reverse (Reagents and Tools Table) and cloned into pEGFP-C1 (Clontech Laboratories, Inc., CA, USA) digested with EcoRI and NotI to express N-terminally EGFP-tagged H3.1 (EGFP-H3.1) [plasmid collection (pc) number: pc2099, Addgene ID: 241429]. To create a site directed point mutation from lysine to methionine at lysine 36 (AAG → ATG; K → M), a DNA polymerase reaction (ALLin™ HiFi DNA Polymerase, Catalog no:.HLE0201, highQu GmbH, Germany) was performed using EGFP-H3.1 plasmid (pc2099) with the primers (H3.1 K36M Forward and H3.1 K36M Reverse), followed by a KLD (Kinase, Ligase, DpnI) reaction with two iterations of 25 °C for 30 min, and 37 °C for 30 min (Reagents and Tools Table). The product was transformed, and multiple clones were selected and cultured. Then, gDNA was isolated to verify the mutation in plasmids by DNA sequencing. One plasmid clone with EGFP-H3.1 with lysine 36 to methionine (H3.1 K36M) was used for the experiments [plasmid collection (pc) number: pc5139, Addgene ID: 241430].

To target the H3.1 with K36 or M36 to the PCH, mESC J1 cells were transiently transfected with EGFP-H3.1 (K36 or M36), and an expression vector encoding the sequence of the GFP-binding $V_H H$ domain (GBP) fused to MajSat (pc2469) (Casas-Delucchi et al, 2012a; Rothbauer et al, 2008). Transient transfections were performed using nucleofection (Amaxa NucleoFector II, Lonza Ltd., Basel, Switzerland) with 2 µg of pGBP MajSat (pc2469) and 1 µg of H3.1 (pc2099 or pc5139). For live-cell experiments, cells were also transfected with 2 µg of a plasmid coding for mRFP-tagged PCNA (pc1054) (Sporbert et al, 2005). The transfected cells were seeded on sterilized coverslips for 16 h (one cell cycle duration to cover at least one round of H3.1 incorporation) before starting the experiments.

## Western blotting

The Antibodies section of the Reagents and Tools Table lists all the antibodies and dilutions used.

The histone proteins were acid-extracted using this protocol (Shechter et al, 2007). Around $10^6$ cells were taken from each cell type and centrifuged. The cells were washed with 1x PBS before resuspending in 1 ml of hypotonic lysis buffer (10 mM Tris-Cl pH

8.0, 1 mM KCl, 1.5 mM $MgCl_2$, and 1 mM DTT). Protease inhibitors were supplemented before use. The cells were incubated for 30 min on a rotator for hypotonic swelling and lysis by mechanical shearing. The nuclei were pelleted using centrifugation at $10,000 \times g$ for 10 min at 4 °C. The nuclei were resuspended in 0.4 N $H_2SO_4$ and incubated overnight. Care was taken to resuspend the clump very well. The nuclei debris were removed by centrifuging the mixture at $16,000 \times g$ at 10 min, and the supernatant was transferred to a new tube. Around 132 ul of 100% TCA (Carl Roth, Karlsruhe, Germany) was added dropwise to make a final concentration of 33% and inverted several times to precipitate the histones. The samples were incubated on ice for 30 min. The proteins were pelleted at $16,000 \times g$ for 10 min at 4 °C. The supernatant was removed, and the pellet was washed twice with ice-cold acetone to remove the acid. After drying the histones at room temperature for 20 min, they were dissolved in 50 µl of $ddH_2O$. The concentration of the proteins was measured using Pierce™ 660 nm Protein Assay Reagent (Catalog no:.22660, Thermo Fisher Scientific, Waltham, MA, USA). From each condition, 4 µg of proteins were denatured in 6x protein loading dye (560 mM Tris-HCl, pH 6.8, 60 mM DTT, 6 mM EDTA, 30% glycerol, and 0.6% bromophenol blue) for five minutes at 95 °C and were loaded in each well to run on a 15% SDS-PAGE. The gel was transferred onto 0.45-µm nitrocellulose membranes (GE Healthcare/Whatman; Catalog #10600002) and imaged using an AI600 imager (GE Healthcare) before blocking the membranes with 3% low-fat milk in PBS 1× for 1 h. The primary antibody against H3K36me3 (diluted 1:10,000) and H3 (diluted 1:2000) was diluted in the blocking buffer and incubated overnight at 4 °C. After washing with 0.01% Tween-20 in PBS 1×, the membrane was incubated in the following secondary antibodies: anti-rabbit IgG Cy3 (1:1000), and anti-rat IgG Cy5 (1:1100). After washing thrice with 0.01% Tween-20 in PBS 1×, the images were taken in the AI600 imager.

## Genome replication labeling, visualization, and immunostaining

The cells seeded on sterilized coverslips with respective media for the replication labeling and visualization experiments were pulse-labeled with 10 µM of EdU for 10 min before washing with PBS 1× and fixing with 3.7% formaldehyde (Carl Roth, Karlsruhe, Germany) in PBS 1× for 10 min for a single pulse detection. For a pulse-chase experiment to infer the replication progression, after the first pulse using 10 µM of EdU (Cat.No.: 7845.1, Click-IT-EdU cell proliferation assay, Carl Roth, Karlsruhe, Germany) for 10 min, the cells were washed twice with the appropriate warm media supplemented with 50 µM of thymidine for 10 min (Cat.No.: T1895, Sigma-Aldrich Chemie GmbH, Taufkirchen, Germany) to stop the incorporation of EdU. The cells were washed twice with PBS 1× and incubated in fresh media for another 3 h. The cells were washed with PBS 1× before fixing with 3.7% formaldehyde (Cat.No.: F8775, Merck KGaA, Darmstadt, Germany) in PBS 1× at room temperature for 10 min. After fixation, the cells were washed thrice with PBS 1×.

Unless otherwise mentioned, all the immunostaining was performed inside a dark, humidified chamber at room temperature. First, the cells were permeabilized with 0.5% Triton X-100 (Carl Roth, Karlsruhe, Germany) in PBS 1× for 10 min, followed by three washes with 0.05% Tween in PBS 1×. To give access to the PCNA

epitope, the cells were incubated with ice-cold methanol for 10 min. The cells were again washed thrice with a washing buffer (0.05% Tween in PBS 1×) and blocked with blocking buffer (4% BSA and 1% fish skin gelatin in PBS 1×) for 30 min.

For the detection of EdU, cells were incubated in Click-IT cocktail mix of 100 mM Tris-HCl pH 8.5, 10 mM CuSO₄, 1 µM 5-TAMRA-Azide (Cat.No.: CLK-FA008-1, Jena Bioscience, Jena, Germany), and 100 mM ascorbic acid diluted in water for 30 min. Cells were washed thrice with 0.05% Tween in PBS 1× for 5 min. To detect the histone modifications or nuclear proteins, respective primary antibodies were diluted in the blocking buffer and incubated overnight at 4 °C. The cells were washed five times and incubated in suitable secondary antibodies for 1 h before washing five times with the washing buffer. All the cells were stained with 10 mg/mL DAPI (4′,6-diamidino-2-phenylindole, Cat.No.: D9542, Sigma-Aldrich Chemie GmbH, Steinheim, Germany) for 10 min and mounted on Vectashield (Cat.No.: VEC-H-1000, Vector Laboratories Inc., Burlingame, CA, USA). All the coverslips were sealed with transparent nail polish and air-dried in the dark.

## RNA preparation, reverse transcription, and quantitative PCR (qPCR) analysis

The total RNA from the control (ES J1 +esiRNA_GFP) and KMT knockdown cell samples (ES J1 +esiRNA_SETD2 +esiR-NA_NSD1) was extracted using the RNeasy Mini kit (Cat. No.: 74104, QIAGEN GmbH, Hilden, Germany) following the manufacturer's instructions. The cDNA was generated by reverse transcribing the total RNA using the SuperScript™ IV First-Strand Synthesis System (Cat.No.: 18091050, Invitrogen, USA) with random hexamers. The transcript levels of major satellites were quantified by qPCR using the StepOnePlus Real-Time PCR System (Applied Biosystems) with SYBR premix (Platinum™ SYBR™ Green qPCR SuperMix, Cat. No.: 11744100, Thermo Fisher Scientific, MA, USA). 100 ng total cDNA was applied for each reaction. The reactions were run with the following profile: incubation at 50 °C for 2 min, denaturation at 95 °C for 2 min, followed by 45 cycles of denaturation at 95 °C for 15 s, annealing at 55 °C for 30 s, and extension at 72 °C for 30 s. The qPCR was performed with three technical replicates for each target.

The primer pairs for major satellite were; qPCR MajSat Forward, and qPCR MajSat Forward and for GAPDH control were qPCR GAPDH Forward, and qPCR GAPDH Reverse (Reagents and Tools Table).

## RNA FISH

To perform RNA FISH, the cells were pre-extracted with ice-cold 0.5% Triton X-100/PBS 1× supplemented with 10 mM Vanadyl Ribonucleoside Complex (VRC, Catalog No.: S1402S, New England Biolabs, MA, USA) for 5 min on ice, removed and fixed immediately with 3.7% formaldehyde in PBS 1× for 10 min. The cells were further washed with PBS 1× and briefly permeabilized with 0.5% Triton X-100 supplemented with 10 mM VRC for 10 min. After washing with PBS 1×, cells were dehydrated with 70, 80, and 100% for 2 min each and air-dried. The LNA probes were prepared with 0.5 µM fluorescently labeled LNA probes for both forward and reverse in 50% formamide (Catalog No.: D4551,

Sigma-Aldrich Chemie GmbH, Steinheim, Germany), 5 µg/ml fish sperm DNA (Catalog No.:11467140001, Roche, Basel, Switzerland), SSC 2×, 10 mM VRC, and 2 mg/ml BSA. We used a mix of both LNA probes conjugated with FAM or TET to detect the forward (major 1 and major 3) and/or reverse (major 2 and major 4) MajSat RNA (Reagents and Tools Table) (Probst et al, 2010).

## Fluorescence recovery after photobleaching (FRAP)

FRAP imaging and bleaching experiments were performed using a Leica TCS SP5II confocal laser scanning microscope (Leica Microsystems, Wetzlar, Germany) equipped with an oil immersion Plan-Apochromat ×100/1.44 NA objective lens (pixel size in XY set to 76 nm) and laser lines at 405, 488, 561, and 633 nm. All imaging was conducted in a closed live-cell microscopy chamber (ACU, Olympus) at 37 °C with 5% CO₂ and 60% humidity, mounted on the Leica TCS SP5II microscope. The emission of GFP was captured using the detection range 495–549. For standard bleaching microirradiation, a preselected spot (1-µm diameter) within a chromocenter in the nucleus was microirradiated for 1.5 s with the laser line 488 nm laser set to 100%. Before and after bleaching, confocal image series of one mid-stack z-section of the nucleus were recorded in 10 s intervals, for 1 min before the bleaching and for 10 min after the bleaching. The following conditions were used: MajSat-GFP (as negative control) (pc1803), PCH-targeted H3.1 K36 or H3.1 K36M (Lindhout et al, 2007).

All analysis steps for the confocal microscopy images from FRAP experiments were performed using ImageJ (Schindelin et al, 2012; Schneider et al, 2012). Images were first corrected for cell movement, and subsequently mean intensity of the bleached region was divided by the mean intensity of the whole nucleus (both corrected for background) using ImageJ software. For each experimental condition, at least 15 cells were used. FRAP data were normalized by pre-bleach fluorescence intensity. All fits were performed on averaged normalized FRAP curves, and the resulting fit parameters are reported as the mean ± standard error of mean (SEM) for two or three independent experiments. Curve fitting was done to single out the exponential equation.

## Microscopy image acquisition

High-throughput image acquisition was performed on a Nikon CREST system with a 20x (SPlan Fluor LWD DIC, air, NA: 0.7) or a 40X (Plan Apo λ DIC, air, NA: 0.95) mounted on a TiE2 platform (Nikon Instruments Inc., Japan). The background was defined by imaging negative staining (for EdU: EdU Click-IT staining without EdU incorporation, secondary antibody staining without primary antibody staining, and DAPI cells imaged without DAPI staining) directly on the Nikon NiS software.

High-resolution 3D images were acquired on an LSM 980 microscope with Airyscan 2 (Carl Zeiss AG, Oberkochen, Germany) with a 63x C plan-apochromatic objective (NA: 1.4) or 40x plan-apochromatic (NA: 0.95). The hyperstack images were acquired in 3D using Airyscan SR mode. The voxel size of the images was kept the same whenever a direct comparison was performed. The acquired images were further processed with the Airyscan joint deconvolution using the default setting to generate super-resolved images on the Zen software.

## Image analysis

The Reagents and Tools Table lists the image and data analysis software and associated plugins/packages in the Software section. All image analyses were performed on the FiJi platform except for the chromatin compaction analysis, which was conducted on R as earlier (Pradhan et al, 2024). All the scripts used are uploaded here. Before analysis, all airyscan images were converted to 8-bit. tiff format using ASJD_16_to_8_bit.ijm, except for the chromatin compaction analysis.

High-throughput image analysis: The StarDist plugin (FiJi) was used to segment the nuclei using the DAPI channel, and nuclear masks were saved (see script StarDistMacro.ijm). The measurement was manually set in the FiJi before measurement or using the set measurement command in the macro. The nuclear masks were used to measure total DAPI intensity and mean EdU intensity, along with respective total histone modification intensity, using the ROI manager (see script: ROI_Image_Measure.ijm). The total RNA FISH intensity was also measured using this approach. These values were exported to R for further analysis. To segment the pericentromeric heterochromatin (PCH), a Gaussian blur (sigma = 1) was applied to the DAPI channel or GFP (H3.1 targeted experiments), and a threshold was used to segment the DAPI-dense regions. Signal intensity from other channels (histone modification or RNA FISH) was measured using the segmented mask.

High-resolution image analysis: Prior to all image analysis, all the channels were separated, and 3D nuclei were segmented using the DAPI channel to define the nuclear region of interest (ROI). First, the 3D stack of DAPI was processed with a Gaussian filter (pixel radius = 2) and normalized (process > enhance contrast > saturated pixels = 0). The nucleus was segmented with 3D nuclei segmentation in the 3D suite plugin, a binary image was created, and further processed with dilations, fill holes, and erode (see script "Batch_DAPI_Segmentation_Process_3D.ijm"). This ROI was applied to all the channels using "image calculator", applying "min" between the DAPI mask and the channel of interest (see script "Image_Calculator_Min_Stack.ijm"). All DAPI masks were manually cross-checked, as most of the mESCs nuclei are irregular in shape, leading to suboptimal DAPI segmentation.

The DAPI-masked images were used further to segment subnuclear structures. The DNA-dense PCH was segmented by applying a Gaussian blur, followed by a threshold and size-based segmentation using the simple segmentation option of the 3D suite plugin. In the case of the PCH-targeted EGFP-H3.1, the masked GFP channel was used to segment the PCH using the same approach (CC_Seg_DAPI_GFPMajSat.ijm).

The 3D spot segmentation was performed using a combined approach of threshold and cluster-based segmentation. First, the 3D stack was processed using a mean filter (pixel radius = 1) and normalized. The FindStackMaxima Macro (https://imagej.nih.gov/ij/macros/FindStackMaxima.txt) was used to find all the local maxima. The corresponding voxels of 3D Maxima were extracted from the processed image using the image calculator, which was used as a seed ("Image_Calculator_Min_Stack.ijm"). The seed and processed images were used to segment the spots (e.g., H3K36me3) in 3D spot segmentation (3D suite) using a Gaussian fit to determine the intensity value

used as the threshold to stop the voxel clustering (see script: Batch_3D_Spot_Segmentation). The segmented objects were further processed with a 3D watershed split (3D suite) to separate the closely clustered spots. The objects were imported into 3D ROIManager (3D suite) for further quantifications and measurements.

To quantify the number of RNA pol II spots inside the PCH, the 3D ROI manager was used with the script "Counting_Spots_within_ROI.ijm". The JACOP plugin was used on Fiji using the script (JACOP_On_Seg_Obj.ijm) to measure the 3D colocalization between two segmented channels.

Chromatin compaction analysis: Only 16-bit super-resolved images were analyzed using the "Nucim" library available on the platform "R" to subdivide voxels inside individual nuclei into chromatin compaction classes and mapping signals from other channels to individual compaction classes (see script Nucim_Analysis_Script.R). First, the DAPI channel was segmented and used to mask the region of interest. Individual voxels within a single nucleus were assigned to a specific compaction class based on the probability of this voxel belonging to the same class computed from a hidden Markov random field (HMRF) stochastic model, classifying the nuclei into seven different compaction classes, where the first four classes represent active nuclear compartment (ANC), and last three classes represent chromatin inactive nuclear compartment (INC). Spots from other channels were further mapped into these subclasses based on a combined threshold and intensity method, where first the spots were segmented with the threshold method, followed by an intensity-weighted calculation of the relative fraction, leading to more intense signals having a larger impact and low-intensity signals having less impact.

## Statistical analysis and data visualization

All datasets were processed and analyzed in R (tidyverse, ggplot2), and significance tests were performed with either ANOVA with Tukey's HSD test (multiple variables) or pairwise Wilcoxon/t-test. In some instances, Cliff's Delta non-parametric effect size was measured. The visualization was performed using ggplot2 and Adobe Illustrator.

After performing the chromatin compaction classification and mapping colors to classes, the datasets were imported to Prism for visualization.

# Data availability

The datasets for each figure and the scripts used for image analysis are uploaded to: https://doi.org/10.48328/tudatalib-1681.4.

The source data of this paper are collected in the following database record: biostudies:S-SCDT-10_1038-S44319-025-00575-6.

# Peer review information

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

## Acknowledgements

We are indebted to Anne Lehmkuhl and Diana Imblan for their technical support. We thank Hector Romero, Maruthi Kumar Pabba and Cathia Rausch for their help and comments on the project. We also thank Songtao Jia for his comments. This research was funded by the Deutsche Forschungsgemeinschaft (DFG, German Research Foundation)—Project ID 393547839—SFB 1361, and CA 198/9-2 Project ID 232488461 to M.C.C.

## Author contributions

**Sunil Kumar Pradhan**: Conceptualization; Formal analysis; Investigation; Methodology; Writing—original draft; Writing—review and editing. **Hui Zhang**: Formal analysis; Investigation; Methodology; Writing—review and editing. **Ksenia G Kolobynina**: Formal analysis; Investigation; Methodology; Writing—review and editing. **Alexander Rapp**: Investigation; Methodology. **Maria**

**Arroyo**: Formal analysis; Investigation; Methodology; Writing—review and editing. **M Cristina Cardoso**: Conceptualization; Resources; Supervision; Funding acquisition; Writing—review and editing.

Source data underlying figure panels in this paper may have individual authorship assigned. Where available, figure panel/source data authorship is listed in the following database record: biostudies:S-SCDT-10_1038-S44319-025-00575-6.

## Funding

## Disclosure and competing interests statement
The authors declare no competing interests.

# Expanded View Figures

**Figure EV1. Cell cycle-dependent dynamics of the histone modification levels.**

(A) Asynchronously growing cell populations pulsed with EdU are stained for EdU and histone modifications. After high-throughput imaging, the images were imported to FiJi, and the StarDist plugin was used to segment nuclei, followed by intensity measurement of DNA content, EdU, and histone modifications. The cell populations are sorted into (non)replicating using the mean EdU intensity; the non-replicating cells are divided into G1 and G2 based on the DNA content from the DAPI sum intensity. The example boxplot shows the corresponding histone modifications sum intensities normalized to DNA content and further normalized to G1 to see the fold change. The p-values inferred from ANOVA followed by Tukey's honest significant difference test are: G1:S = 0.004971475, S: G2 = 3.593188e-33, G1:G2 = 3.149946e-12. $n = 4730$ (three biological replicates). The boxplot shows the median (line), interquartile range (box: 25th to 75th percentile), and whiskers extending to data within 1.5× the IQR. n.s. not significant; *$p < 0.05$; **$p < 0.005$; ***$p < 0.0005$. (B) The plot shows the fold changes (normalized to G1) in the normalized histone modification intensity across different cell cycle phases for various histone modifications in mouse embryonic stem cells (mESCs). Statistical significance was assessed using a pairwise $t$-test (all $p$ values are provided in the source data files). Asterisks indicate significant differences in normalized histone modification intensity between the specified cell cycle phases: n.s. no significant, *$p < 0.05$, **$p < 0.05$, and ***$p < 0.005$. In this figure, statistical significance is represented as * for any pairwise comparison where the difference is significant, and n.s. for not significant. Three independent biological replicates were pulled together after cell cycle classification for each histone modification. $n = 59,934$ from all histone modifications. (C) A table summarizes the significantly changed histone modification levels through the cell cycle. The amounts of the transcription, enhancer, and nascent chromatin-associated histone modification marks are increased in the S phase, whereas heterochromatin marks are decreased. Not all the histone modifications are recovered post-replication. Source data are available online for this figure.

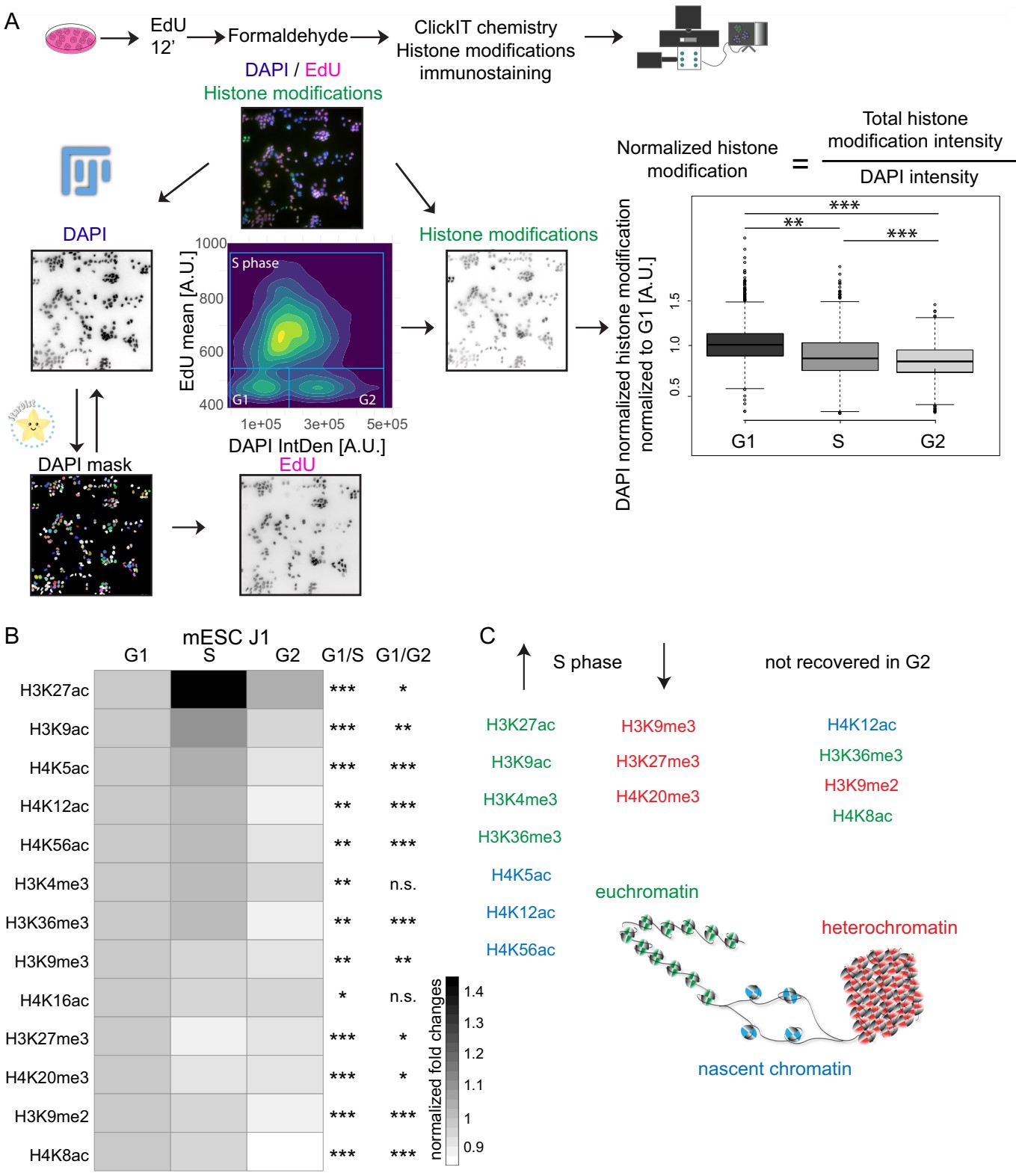

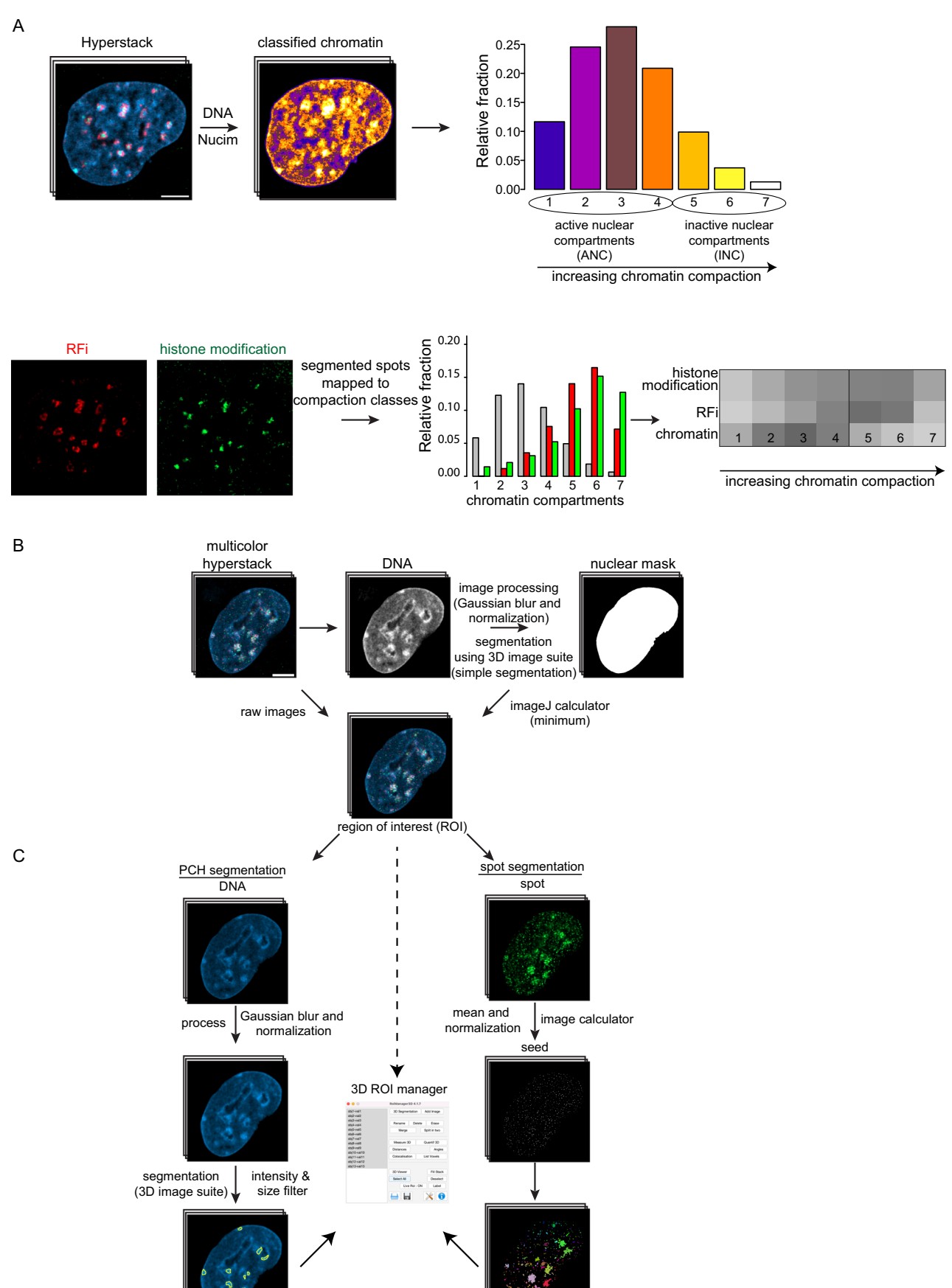

◀  **Figure EV2.  High-resolution image analysis pipelines.**

(**A**) The pipeline shows the approach for chromatin compaction classification and mapping RFi/histone modification to different compaction classes using the statistical image analysis tool Nucim on the R platform. Individual voxels inside the nuclei are assigned to different compaction classes (increasing compaction from 1 to 7), where the first four represent active nuclear compartments, and the last three classes represent inactive nuclear compartments. The relative fraction of RFi/histone modifications in each chromatin class is measured. (**B**) The pipeline shows the approach to segmenting the nuclei and defining the region of the nuclei. The DAPI channel (DNA) was first processed with a Gaussian blur and was normalized using the stack histogram. Using the 3D ImageJ suite plugin, the processed image was segmented (simple segmentation, with size filter). The size selection removes small debris that might be present outside the nuclei. The 3D mask was binarized, and the image calculator (minimum) operation was used between the DNA mask and the channel of interest (including original DNA) to remove the signals outside the nuclei. (**C**) The scheme shows approaches to segment the pericentromeric heterochromatin (PCH) or spots (e.g., histone modifications/RFi). The PCH was segmented using a high intensity and volume threshold filtering in only the DAPI-dense regions. For RFi/RNA Pol II/histone modifications, a combined approach of 3D local maxima and intensity was used. First, the 3D local maxima of the images were extracted and used as seeds, around which an intensity threshold was applied to segment the 3D spots. For spot segmentation, the 3D ImageJ suite was used (spot segmentation, local threshold method = Gaussian fit, watershed = yes). All the masks were imported into the 3D ROI manager for further quantification/colocalization measurements. Scale bar: 5 μm.

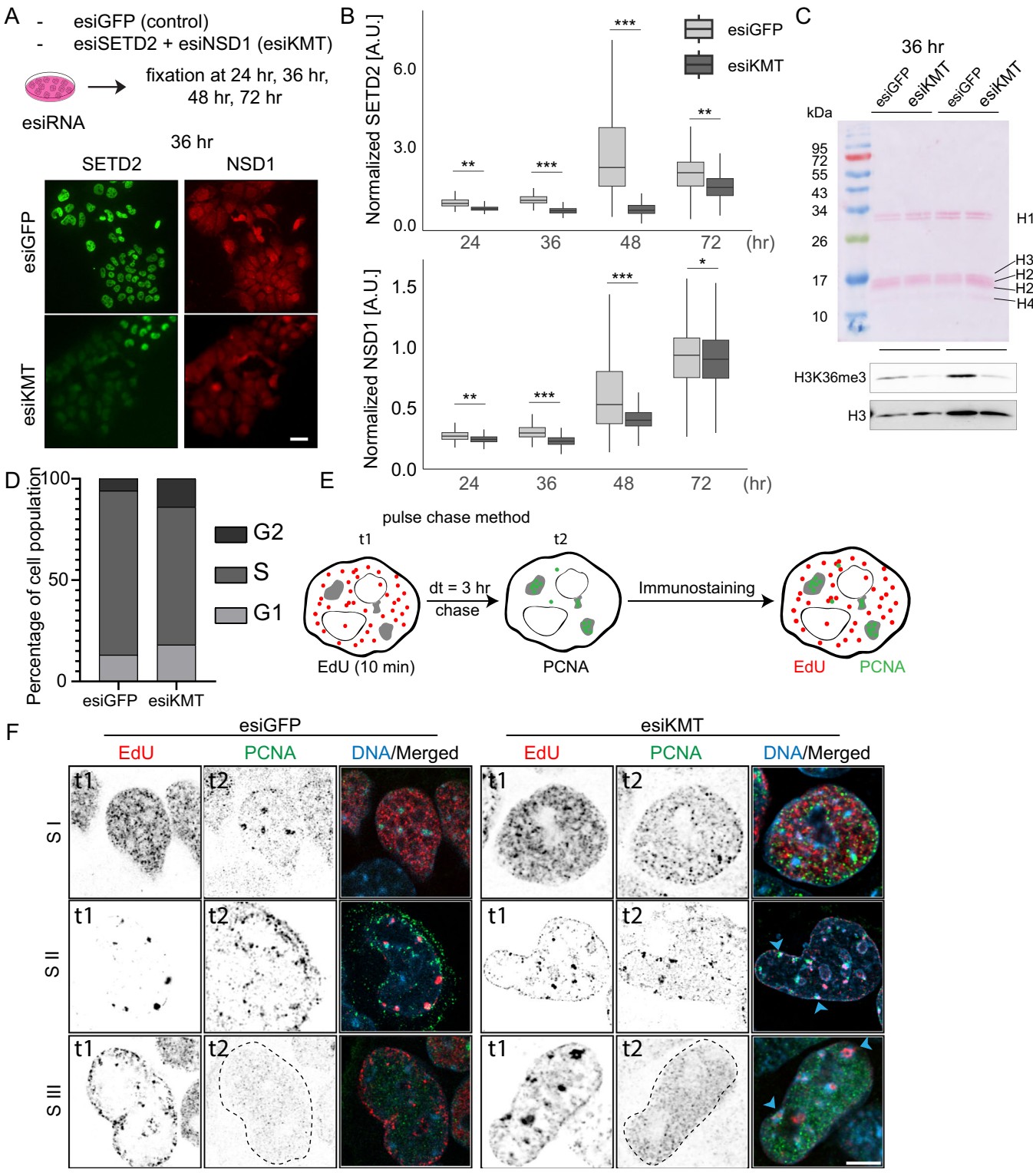

◄ **Figure EV3. Knockdown strategy to reduce H3K36me3 using RNA interference, and its effect on replication timing.**

(A) The knockdown of two K36-specific methyltransferases (KMT), SETD2 and NSD1, was performed using esiRNA, and the level was quantified at different time points using high-throughput imaging and analysis. Example images show the reduced levels of SETD2 and NSD1 in the nucleus. Scale bar:10 µm. (B) A significant reduction of SETD2 and NSD1 levels was observed from 36 to 48 h, while the effect was less prominent around 72 h. $n$ (eGFP) = 6021, $n$ (SETD2 + NSD1) = 6212. Cell images were used from three biological replicates. Relevant $p$ values are inferred from pairwise Wilcoxon test between esiGFP and esiKMT: SETD2 (24 h: 0.00091, 36 h: 0.00021, 48 h: 0.00019, 72 h: 0.00058), NSD1 (24 h:0.00073, 36 h: 0.00042, 48 h: 0.00032, 72 h: 0.00059). The boxplot shows the median (line), interquartile range (box: 25th to 75th percentile), and whiskers extending to data within 1.5× the IQR. Statistical significance was performed using the Wilcoxon test (not significant (n.s.) is given for $p$ values ≥0.05; one star (*) for $p$ values <0.05 and ≥0.005; two stars (**) is given for values <0.005 and ≥0.0005, and 0.0005 to 0 are given (***). (C) The knockdown of H3K36me3 was validated using acid-extracted histones between GFP and KMT knockdown samples. The ponceau (upper blot) shows the acid-extracted histone. On the lower panels blot shows the detected histone H3 along with H3K36me3. (D) The plot shows the percentage of the S-phase population among esiGFP and esiKMT measured from high-throughput image analysis using EdU and DNA content. Upon knockdown of KMTs, a relatively lower population was detected in the S phase. $n$ (esiGFP) = 1202, $n$ (esiKMT) = 1461, merged from two biological replicates. (E) The illustration shows the pulse-chase method to capture the order of RFi patterns to infer replication timing. S phase spatial patterns S I–S III were inferred based on EdU incorporation and PCNA spatial patterns: S I (EdU-negative, PCNA-positive), S II (early EdU labeling with characteristic PCNA distribution), and S III (EdU-positive with loss of punctate PCNA foci). (F) Images show the results of a pulse-chase experiment to capture the spatiotemporal dynamics of genome replication. In the control, the replication progresses from euchromatin to constitutive heterochromatin, followed by lamin-associated domains. In KMT, knockdown, the pericentromeric heterochromatin (marked by blue arrow), and lamin-associated domains are replicated concomitantly after euchromatin is replicated, $n$ (esiGFP) = 12, $n$ (esiKMT) = 14. Scale bar: 5 µm. Source data are available online for this figure.

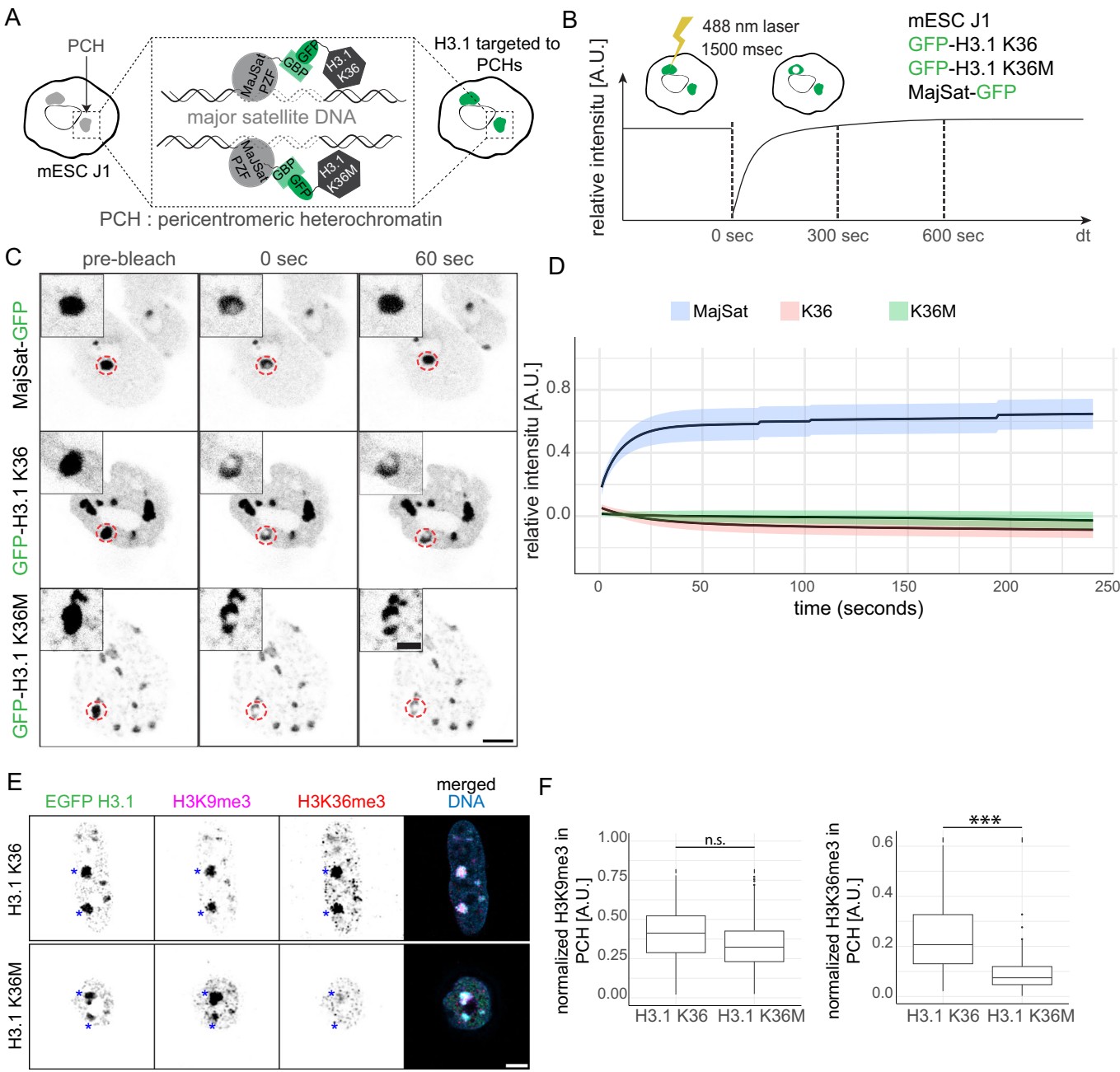

Figure EV4. Knockdown strategy to reduce H3K36me3 from pericentromeric heterochromatin (PCH) using oncohistone H3.1 K36M.

(A) The illustration shows the targeting strategy of H3.1 or H3.1 K36M tagged with GFP to the PCH using PZF-MajSat nanobody fusion protein to reduce the level of H3K36me3 locally from the PCH by taking advantage of the replication-dependent H3.1 incorporation. (B) The illustration shows the fluorescence recovery after photobleaching (FRAP) strategy to gauge the dynamics of H3.1 after incorporation. (C) Images show the pre-bleach and post-bleach dynamics of H3.1 (H3.1 K36 and H3.1 K36M), with MajSat-GFP as a negative control. See also Movie EV1. Movie EV1 shows the live-cell imaging of FRAP experiments with corresponding plots. The affected chromocenter is highlighted by the dashed red circle on the whole-nucleus view and the zoomed image attached to the whole-nucleus view. Scale bar: 5 and 2 μm in the zoomed image. (D) The plot shows the fluorescence recovery after photobleaching curves displayed as mean ± standard error of mean (SEM) for two biological replicates. n (MajSat-GFP) = 16, n (GFP-H3.1 K36) = 16, and n (GFP- H3.1 K36M) = 16. (E) Images show the immunofluorescence detection of H3K36me3 and H3K9me3 in the H3.1 GFP-targeted cells from the G1/early S phase fraction. Only the cells with a relatively higher intensity of GFP-H3.1 in the PCH and lower intensity elsewhere in the nucleus were inferred as targeted and selected for analysis. (* marks the PCH). Scale bar: 5 μm. (F) The normalized (to DNA) total intensity of H3K9me3 and H3K36me3 was plotted for both conditions using segmented GFP masks (of PCH) in each nuclei. Both pairwise Wilcoxon and Cliff's delta effect size were performed. For H3K9me3, p value = 0.118 (Cliff's Delta = 0.240). For H3K36me3, p = 0.000011 (Cliff's Delta = 0.671). Sample sizes (number of nuclei from three biological replicates) are n = 18 for H3.1 K36 and n = 19 for H3.1 K36M. The boxplot shows the median (line), interquartile range (box: 25th to 75th percentile), and whiskers extending to data within 1.5× the IQR. Statistical significance was performed using the Wilcoxon test (not significant (n.s.) is given for p values ≥0.05; one star (*) for p values <0.05 and ≥0.005; two stars (**) is given for values <0.005 and ≥0.0005, and 0.0005 to 0 are given (***). Source data are available online for this figure.

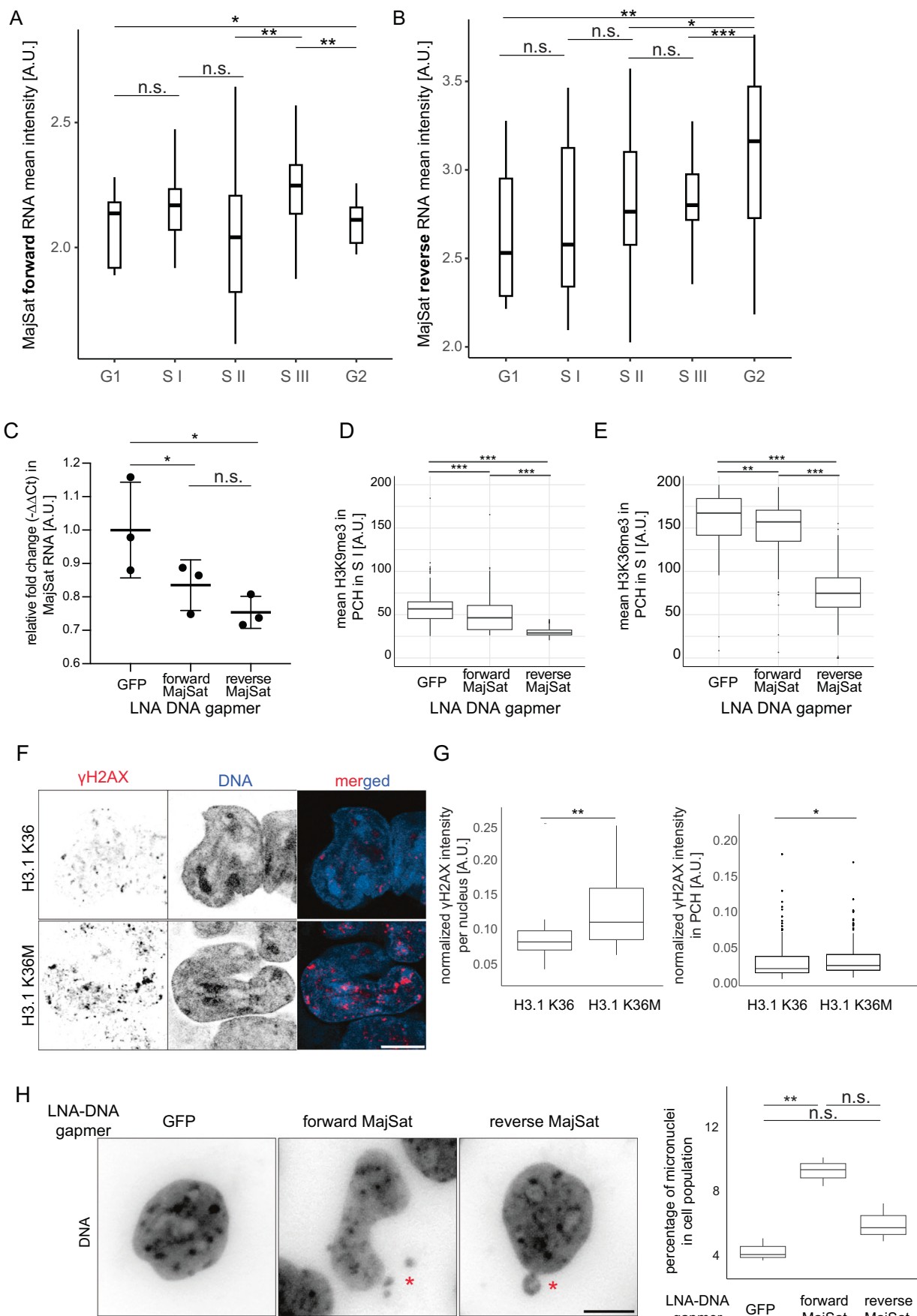

◀ **Figure EV5. Cell cycle-dependent MajSat RNA expression, its epigenetic modulation, and role in maintaining heterochromatin stability.**

(A) The plot shows the expression of MajSat forward RNA during cell cycle progression inferred from RNA FISH at different cell cycle stages. The statistical test was performed using ANOVA followed by Tukey's honest significant difference (HSD). All p-values are provided in the source data files. The relevant $p$ values are: G1:S I = 0.625911770, S I: S II = 0.040221033, S II: S III = 0.002596268, and S III: G2 = 0.008092236 & G1: G2 = 0.999995863. $n = 170$ from three biological replicates. (B) The plot shows the expression of MajSat reverse RNA during cell cycle progression. The statistical test was performed as above, and $p$ values are provided in the source data. The relevant $p$ values are: G1:S I = 0.96182994, S I: S II = 0.95379238, S II: S III = 0.99423587 and S III: G2 = 0.63173138 and G1: G2 = 0.04847357. $n = 176$ from three biological replicates. For both plots in (A, B), the lower and upper whiskers of the boxplot correspond to the 25th and 75th percentiles, the box to the 50th percentile, and the line depicts the median. (C) Plot shows the relative fold change in the respective MajSat RNA upon locked nucleic acid DNA gapmer-mediated interference. Three independent biological replicates were performed, using a pairwise $t$-test; GFP-Forward MajSat: $p = 0.0130$, GFP-Reverse MajSat: $p = 0.0477$, Forward MajSat-Reverse MajSat = 0.493786605. (D) The plot shows the mean H3K9me3 in the PCH upon LNA-DNA-mediated interference. In general, the H3K9me3 level is reduced upon interference of both forward and reverse MajSat RNA and is particularly affected in the reverse. The statistical test was performed using ANOVA followed by Tukey's honest significant difference (HSD). The $p$ values are: GFP-forward = 0.00001, GFP-reverse = 0.0000014, forward-reverse = 0.0000165. (E) The plot shows the mean H3K36me3 in the PCH upon LNA-DNA-mediated interference. While in both forward and reverse MajSat interference reduced H3K36me3, the level of H3K36me3 was reduced significantly in the reverse. The statistical test was performed using ANOVA followed by Tukey's honest significant difference (HSD). The $p$ values are: GFP-forward = 0.0002552, GFP-reverse = 0.000001, forward-reverse = 0.00001. For both boxplots (D, E), the n(GFP) = 14, n(forward) = 14, and n(reverse) = 12 from two biological replicates. (F) Images show the γH2AX levels upon targeting of H3.1 with/without K36M. (G) The first plot shows the quantification of normalized γH2AX intensity per nucleus, and the second plot shows per PCH area in the H3.1 K36 or H3.1 K36M-targeted cells. The significance test was performed using a pairwise $t$-test. $n$ (H3.1 K36) = 19, $n$ (KMT) = 21, $p$ value from the pairwise Wilcoxon test $p$ (normalized γH2AX intensity per nucleus) = 0.0025, and $p$ (normalized γH2AX intensity in PCH) = 0.007404. (H) Nuclei stained with DAPI in strand-specific interference of the MajSat transcripts. Asterisks (*) mark the micronuclei. The plot shows the percentage of the micronuclei in the cell population upon the interference of MajSat RNA. The significance test was performed using a pairwise $t$-test; significant difference $p$ value (GFP versus forward MajSat) = 0.0034. n(GFP) = 718, n(forward) = 1225, n(reverse) = 739 from three biological replicates. For plots (D–H), the lower and upper whiskers of the boxplot correspond to the 25th and 75th percentiles, the box to the 50th percentile, and the line depicts the median. (not significant (n.s.) is given for $p$ values ≥0.05; one star (*) for $p$ values <0.05 and ≥0.005; two stars (**) is given for values <0.005 and ≥0.0005, and 0.0005 to 0 are given (***); only the significant differences are shown). Scale bar: 5 μm. Source data are available online for this figure.

