## [Peer Review File · EMBO Reports]

Dynamic association of H3K36me3 with pericentromeric heterochromatin regulates its replication time

Sunil Pradhan, Hui Zhang, Ksenia Kolobynina, Alexander Rapp, Maria Arroyo, and M. Cristina Cardoso

Corresponding author(s): M. Cristina Cardoso (cardoso@bio.tu-darmstadt.de)

Review Timeline:

Submission Date:	6th Mar 25
Editorial Decision:	14th Apr 25
Revision Received:	13th Jun 25
Editorial Decision:	28th Jul 25
Revision Received:	21st Aug 25
Accepted:	26th Aug 25

Editor: Achim Breiling

Transaction Report:

Dear Prof. Cardoso,

Thank you for the submission of your manuscript to EMBO reports. I have now received the reports from the three referees that were asked to evaluate your study, which can be found at the end of this email.

As you will see, the referees think that these findings are of interest. However, they have several comments, concerns, and suggestions, indicating that a major revision of the manuscript is necessary to allow publication of the study in EMBO reports. As the reports are below, and all the referee concerns need to be addressed, I will not detail them here.

Given the constructive referee comments, I would like to invite you to revise your manuscript with the understanding that the concerns of the referees must be addressed in the revised manuscript and in a detailed point-by-point response. Acceptance of your manuscript will depend on a positive outcome of a second round of review. It is EMBO reports policy to allow a single round of revision only and acceptance of the manuscript will therefore depend on the completeness of your responses included in the next, final version of the manuscript.

- 1) a .docx formatted version of the final manuscript text (including legends for main figures, EV figures and tables), but without the figures included. Figure legends should be compiled at the end of the manuscript text.
- 2) individual production quality figure files as .eps, .tif, .jpg (one file per figure), of main figures and EV figures. Please upload these as separate, individual files upon re-submission.

- 4) a complete author checklist, which you can download from our author guidelines (<https://www.embopress.org/page/journal/14693178/authorguide>). Please insert page numbers in the checklist to indicate where the requested information can be found in the manuscript. The completed author checklist will also be part of the RPF.

- 5) that primary datasets produced in this study (e.g. RNA-seq, ChIP-seq, structural and array data) are deposited in an

appropriate public database. If no primary datasets have been deposited, please also state this in a dedicated section (e.g. 'No primary datasets have been generated and deposited'), see below.

The accession numbers and database should be listed in a formal "Data Availability" section that follows the model below. This is now mandatory (like the COI statement). Please note that the Data Availability Section is restricted to new primary data that are part of this study. This section is mandatory. As indicated above, if no primary datasets have been deposited, please state this in this section

Data availability

6) We now request the publication of original source data with the aim of making primary data more accessible and transparent to the reader. You will receive a separate email with instructions for providing source data with your revised manuscript, including information how to upload and organize the files.

8) Regarding data quantification and statistics, please make sure that the number "n" for how many independent experiments were performed, their nature (biological versus technical replicates), the bars and error bars (e.g. SEM, SD) and the test used to calculate p-values is indicated in the respective figure legends (also for EV and Appendix figures). Please also check that all the p-values are explained in the legend, and that these fit to those shown in the figure. Please provide statistical testing where applicable. Please avoid the phrase 'independent experiment', but clearly state if these were biological or technical replicates. Please also indicate (e.g. with n.s.) if testing was performed, but the differences are not significant. In case n=2, please show the data as separate datapoints without error bars and statistics. See also: <http://www.embopress.org/page/journal/14693178/authorguide#statisticalanalysis>

9) Please add scale bars of similar style and thickness to microscopic images, using clearly visible black or white bars (depending on the background). Please place these in the lower right corner of the images themselves. Please do not write on or near the bars in the image but define the size in the respective figure legend.

10) Please also note our reference format:

12) We now use CRedit to specify the contributions of each author in the journal submission system. CRedit replaces the author contribution section. Please use the free text box to provide more detailed descriptions and do NOT provide your final manuscript text file with an author contributions section. See also our guide to authors: <https://www.embopress.org/page/journal/14693178/authorguide#authorshippinguidelines>

13) All Materials and Methods need to be described in the main text using our 'Structured Methods' format, which is required for

all research articles. According to this format, the Methods section should include a Reagents and Tools Table (listing key reagents, experimental models, software, and relevant equipment and including their sources and relevant identifiers), uploaded as separate file, and a Methods section in which we encourage the authors to describe their methods using a step-by-step protocol format with bullet points, to facilitate the adoption of the methodologies across labs. More information on how to adhere to this format as well as downloadable templates (.doc) for the Reagents and Tools Table can be found in our author guidelines (section 'Structured Methods'):

Please add all primer- and antibody-related information directly to this table.

14) Please order the sections like this, using (only) these names:

Title page - Abstract - Keywords - Introduction - Results & Discussion - Methods - Data availability section - Acknowledgements (including the funding information) - Disclosure and Competing Interests Statement - References - Figure legends - Expanded View Figure legends

15) Please make sure that all the funding information is also entered into the online submission system and that it is complete and similar to the one in the acknowledgement section of the manuscript text file.

Please note that corresponding authors are required to supply an ORCID ID upon submission of a revised manuscript and an institutional e-mail address. Please find instructions on how to link the ORCID ID to the account in our manuscript tracking system in our Author guidelines: <http://www.embopress.org/page/journal/14693178/authorguide#authorshipguidelines>

I look forward to seeing a revised form of your manuscript when it is ready.

Yours sincerely,

Referee #1:

This paper was a pleasure to read. The rationale for each experiment is clearly articulated, the controls for each experiment are present, the approaches are innovative and the conclusions are restricted to what is revealed in the data. Pathways regulating replication timing and the spatio-temporal organization of replication foci are very poorly understood, so this work provides an entryway into one of those pathways. My only struggle with this manuscript came in the last paragraph of the results section. I think we need a few more words to understand why global KD of H3K36me3 leads to a reduction in global H2A.X while local disruption leads to an increase. We have known that aberrations in PCH lead to genome instability for a long time, so that is expected anyhow, but if the authors feel the need to include this final experiment in EV5, the pathway by which global disruption has an apparently opposite effect should be more clearly articulated since all of their other data shows that it will be disrupted at PCH.

Minor comments:

- Something to consider to increase the visibility of this paper: add "through major satellite RNA" to the end of the title.
- When discussing "aberrations" in the RT of heterochromatin, my impression is that these are all RT delays - if correct, why not be more specific and say "leads to delays". Are there cases of advancement of RT?

Referee #2:

In this study, Pradhan and colleagues investigated the role of H3K36me3 at pericentric heterochromatin (PCH) in mouse embryonic stem cells (mESCs) during replication. They analyzed the presence of various histone post-translational modifications (PTMs) at PCH throughout the cell cycle and found that H3K36me3 exhibits a dynamic association. To dissect the role of H3K36me3 at PCH, the authors employed several strategies to reduce H3K36me3 levels, including esiRNA knockdown of H3K36 methyltransferases (such as Setd2 and Nsd1). The authors also co-transfect a GFP-tagged histone H3.1 with K-to-M mutation of residue 36 (H3.1K36M) and a GBP-fused MajSat binding protein, thereby targeting the H3K36M to the PCH. The authors argue that these interventions led to defects in replication timing of PCH and altered major satellite (MajSat) expression, particularly affecting S-phase sub-stages SII and SIII. Notably, treatment with locked nucleic acid (LNA) DNA gapmers against the forward but not reverse MajSat transcripts, presumably blocking/removing MajSat transcript(ion) partially mimicked the H3K36me3 reduction phenotype, supporting a model in which H3K36me3 promotes MajSat transcription, which in turn is necessary for timely replication of heterochromatin and maintenance of genome stability. This reviewer has several concerns, related to the underlying mechanism, the explanation of experimental procedures, the clarity of the text, the data visualization in the figures and the data interpretation.

Major concerns:

- 1) How did the separation of cells based on the replication foci (RFi) occur? Please show several examples in the supplementary. Also, please show the masks and examples of segmented RFi. The same applies to the segmentation of PCH and of other items.
- 2) In order to subdivide the S phase cells the authors used the spatial pattern of RFi (EdU) to segregate SI, SII and SIII substages as shown in Fig 1A. How is the spatial pattern calculated to assign a cell to a given group?
- 3) Fig. 1B. The authors introduce seven different chromatin compaction classes. If the reviewer understands correctly, the 1 -> 7 direction corresponds to the grades of chromatin compaction, but this should be explained, e.g. grade 7 corresponds to the most condensed. The basis for this classification is unclear for this reviewer. Please clarify. In the context of the main topic of the manuscript, is the presentation of this data really needed?
- 4) Authors used the EdU pulse chase followed by PCNA staining to monitor the progression of the S phase cells. However, when they assign the cells to an initial S substage based on EdU this should follow the same criteria between a control and treated cell. There are multiple examples where the assignment of a cell based on the EdU pattern (as shown in Fig 1A) may be inappropriate. For example:
In figure EV3 C: the SIII cell of esiKMT has an EdU spatial pattern which resembles the SII pattern.
In figure 2 E: the SII cell of H3.1 K36M has an EdU spatial pattern which resembles the SI pattern.
In figure 4C: the representative EdU patterns for SI and SII look very similar.
If the assignment of a cell to an EdU pattern has been done manually the reviewer is concerned about experimental biases. Moreover, the applied experimental perturbations may affect replication of some but not other regions of the genome. As a consequence, new combinations of EdU incorporation and PCNA direct labeling patterns may arise. Reviewing the provided images, this reviewer interprets the findings that replication of PCH may be extended, therefore possibly occurring concurrently with replication of perinuclear regions. Hence, an extensive quantification of the data is required which should be provided in the paper. To understand better a possible selective extension of the duration of replication of PCH, the authors may need to consider applying different durations for the chase phase.
- 5) The authors used the above type of pulse chase experiments to argue for a delayed cell cycle S phase. Such experiments should provide a relative estimation, and the authors should provide actual quantification of delayed cell cycle in minutes/hours.
- 6) How do the authors explain the increase of both SETD2 and NSD1 in esiGFP samples from 36 to 48 and further to 72 hours Fig. EV2 B?
- 7) When expressing and recruiting the H3.1K36M protein to the PCH, please show and quantify accumulation of the GFP anchoring protein at PCH. What do the authors think - does the GFP-tagged histone become incorporated into nucleosomes at PCH, or not? If not, how does the H3K36M protein lower the H3K36me3 levels. Does it locally block the activity of the HMTs? Can one detect an accumulation of the SETD2 or NSD proteins at such PCH regions?
- 8) The authors argue that forward strand expression of MajSat transcript is somehow involved in the replication of PCH repetitive sequences and that H3K36me3 is acting downstream. To further extend this model, have the authors observed a change in H3K36me3 staining levels upon treatment of the LNA probes against forward but not reverse MajSat transcripts? What is the impact of LNA probe treatment on MajSat forward and reverse transcript levels in the nuclei? Are the LNA probes affecting replication by removing/lowering the local nuclear concentration of MajSat transcripts or could they directly impair the replication process, and only indirectly affect transcript levels? See also below - are MajSat transcripts actually transcribed just prior or after their own replication or is it occurring at other stages of the cell cycle?
- 9) When is H3K36me3 established at PCH during the cell cycle? Does this relate to MajSat expression? Experimentally, could one abrogate expression and/or H3K36me3 deposition prior to replication to see whether then replication of PCH is affected?
- 10) The authors should provide a proper discussion addressing a) the role of H3K36 demethylases at PCH, given that at least KDM2A was shown to interact with HP1, b) the role of classic PCH histone PTMs such as H3K9me3 and H4K20me3 along with their corresponding methyltransferases (Suv39h1/2, Setdb1, Suv420h1/2) in replication, c) the effect at other heterochromatic components which might overlap with DAPI bright foci, such as telomeric repeats, as well as the d) the similarities and differences of the esiKMT and H3.1K36M PCH targeting experiments.

Minor Comments:

1. The authors show that relative percentages of cells in G1 S and G2 are changed between esiGFP and esiKMT which raises the question of the relative % of SI, SII and SIII population change upon knockdown of KMT or expression of H3.1 K36M or knockdown of maj sat RNA? Please specify.
2. DAPI preferentially binds to AT-rich DNA stretches, present in major satellite repeat sequences. These sequences form the DAPI bright chromocenter foci which show dynamic changes in numbers and sizes during cell cycle and development. Can these changes affect the measured intensity of the DAPI signal and affect the applied normalizations in some way?
3. Can the knockdown of MajSat transcripts verified by FISH or PCR?
4. Is there a dynamic expression of MajSat transcripts during cell cycle and S phase substages? Are the MajSat transcripts RNA expression levels affected by KMT knockdown or expression of H3.1 K36M show a cell cycle specific manner?
5. In figure EV5B, are the micronuclei observed in a particular cell cycle stage or substage of the S phase?
6. Fig. EV3C: Since the analysis of chromatin was based on DAPI staining, please show the DAPI channels accordingly in the same panel. The t1 and t2 labels in this and similar other panels can be removed, as from the description of the pulse and chase method (Methods section), the timing of EdU incorporation and chase duration are clear. This is relevant to other panels organized in the same principle.
7. Results and Discussion, Part 1, Paragraph 1: "However, most of the epigenetic states are re-established, as inferred from the normalized PTM levels in G2, except for H3K9me2, H3K36me3, H4K8ac and H4K12ac (summarized in Fig. EV1C)" : this argument is an oversimplification since only 11 out of 13 PTM show significant changes in G2 compared to G1. Thus, there is only a minority of PTMs which fully reestablish their modification levels after S phase. Please correct.
8. Please explain the rationale for the measurement of MajSat RNA in different cell cultures?
9. It would be a bit easier for the reader if the authors would briefly explain what the advantage of using asynchronous cell culture is, instead of the synchronized, in pulse chasing experiments. I guess, this is to capture all different cells.
10. Given that knockdowns of SETD1 and NSD1 affect H3K36me3 genome wide and part of H3.1 K36M is not at PCH (although the majority is properly targeted), the authors should discuss indirect effects of PCH replication timing effect or in general replication related effects.
11. Results and discussion, paragraph 2, sentence: "based on the spatial pattern of the RFI" - please specify what are the RFI already in this sentence, as this abbreviation has not been introduced before
12. In this sentence: "The tandem repeats of MajSat comprising the PCH are generally transcriptionally incompetent, but these regions are known to potentially be transcribed", the phrasing is a contradictory and not in line with the literature.
13. There are two figures termed as EV4. One is related to the transcription of MajSat in cell lines, and another is about Pol II Ser2 staining upon targeting of H3.1K36M to the PCH. Please correct the labels accordingly in figure legends and in the text.
14. Fig. 2D and Fig EV3B are redundant- the same cartoon appears in both.
15. Fig. 1D. Please report n for each individual stage.
16. The authors should cite literature on why they use gH2A.X levels as a proxy of genome instability
17. Please provide consistent magnifications of the figures for the gH2A.X immunofluorescence data in esiGFP/esiKMT and the H3K36M experiments in figure EV5 A-C.
18. The description of the gH2A.X antibody is not available in the materials and methods, please update the table accordingly.
19. Fig. EV3 B - in the cartoon, why do the red dots disappear in t2? If they correspond to a short pulse of EdU incorporated, they should still be present. According to the cartoon, it may seem that the incorporation of EdU was done for 3 hours, while it was only for 10 mins. Please make the cartoon clearer about the timing of incorporation of EdU, thymidine, and the subsequent fixation of cells.
20. Many sentences can be improved from the perspective of clarity and typos, especially those in figure legends. Examples as follows:
"Briefly, a short pulse of EdU was followed by a chase with an excess thymidine to block further EdU incorporation for three hours and fixed" - please specify what was fixed.

Fig. 1E,F - please correct the legend, as panels E and F are labelled as B and C, accordingly

"Replication timing also influences the epigenome and plays a key role in maintaining the same." Please clarify what is the same?

Referee #3:

In their manuscript titled "Dynamic association of H3K36me3 with pericentromeric heterochromatin promotes its transcription to regulate replication timing" the authors Pradhan et al. delve into understanding the role of the histone post translational modification H3K36me3 in DNA replication and satellite DNA transcription. Specifically, by using asynchronously growing mouse embryonic stem cells, they discovered that H3K36me3 is dynamically regulated at different phases of the cell cycle. In particular, H3K36me3 is upregulated during the G1/S transition and co-localizes with pericentromeric heterochromatin (PCH) prior to the replication of this domain. To test whether H3K36me3 has a role in regulating DNA replication at the PCH, they knocked down the major lysine methyltransferases responsible for H3K36me3 modification to generally reduce H3K36me3. In a parallel approach, they used a sophisticated strategy to specifically target H3K36me3 only at pericentromeric regions using forced integration of a mutated version of H3K36 that cannot be methylated. In both experiments, they observed alteration of chromatin

replication timing, indicating that H3K36me3 may play a role in regulating this process. In addition, they also discovered that upon loss of H3K36me3, the non-coding forward transcript of MajSatDNA is dramatically reduced. By selectively impairing the function of forward MajSatRNA, the authors found similar results to the loss of H3K36me3 - altered replication timing. The authors propose a final speculative model in which H3K36me3 is important for MajSat DNA transcription, with the satellite DNA transcript in turn relevant for chromatin replication timing. The data presented in this manuscript is of good quality, relevant and mostly well-interpreted.

In our opinion, the major conceptual advance in this manuscript is that H3K36 trimethylation is associated MajSat DNA transcription and DNA replication timing. However, we think that other aspects of the manuscript remain underdeveloped, particularly the question of if and how MajSat influences replication timing. In addition, some experiments remain underquantified (see below) while certain conclusions are also not completely supported by data.

Major comments:

1. We found the nuclear compartment classification system (INC/ANC, IC/PR/CDC) very confusing and a schematic is absolutely required in Fig. 1.
2. In general, the microscopy images of the KMT depletion are not convincing, but consistent with the quantifications in EV2B. What was the rationale to not use samples at 48H when the fold change between eGFP control and KMT KD is higher?
3. In the pulse-chase approach to quantify replication timing, the authors use excess thymidine to block further EdU incorporation. Why does replication proceed in the presence of excess thymidine? In general, how this pulse chase experiment works could be explained better.
4. In the experiment where the onco-histone H3.1 K36R is targeted to the MajSat DNA, are H3K36 methylation levels only specifically reduced at the PCH? Do the authors observe that H3K36 methylation at other loci is unaffected?
5. In general, the replication timing data in EV3C, 2E & 4 is not quantified. The role of H3K36me3/MajSat transcription in replication timing is a central conclusion of the manuscript but is only supported by representative images.
6. In Figure 3 (and referred to on page 7), the authors used RNA pol II Ser5 staining to visualize active transcription sites. However, these images are missing from Figure 3, and they need to be shown. In addition, the authors do not fully discuss the implications of this somewhat counterintuitive result.
7. On page 7-8, the authors titled their last result section as, 'H3K36me3-dependent expression of MajSat (forward) maintains the replication timing of constitutive heterochromatin and genome stability'. However, we think that there is not enough evidence to show an active role of MajSat RNA in maintaining chromatin replication timing. First, the gapmers against MatSat RNA have not been validated and whether the knockdown is effective remains unknown. Also, the authors conclude that it is the loss of MajSat forward transcripts that leads to chromatin replication time disruption. However, this could be due to other effects associated with loss of H3K36 methylation. To make their conclusion stronger, the authors could try to perform a rescue experiment in which they introduce forward MajSatRNA in a background in which H3K36me3 is reduced, and see whether this rescues chromatin replication timing. Alternatively, they should interpret their data more carefully.

Minor:

1. There are several instances in the manuscript where the choice of words are unconventional, and therefore imprecise. I've listed two examples below but the manuscript could use copy editing overall.
 - a. Page 3 - '...significant disruption in histone PTM levels....'. Should be 'reduction/depletion' etc.
 - b. Page 4 - '...known to be transcriptionally incompetent,....' Should be 'silent'
2. Abbreviations - RFI is used in page 3 but without indicating what it stands for. Similarly, esiRNA (page 5) should be properly described.
3. On page 5, the authors use H3K9me3 as a negative control for their GFP nano-trap experiment to verify that another H3 post translational modification is not affected. This should be better explained in the text.
4. Figure 2 B, C, p-values and statistical analysis are missing.
5. On page 7, the authors speculate that H3K36me3 could prevent premature transcription initiation (perhaps by keeping POLII paused?) and promoting elongation. Another possibility is that H3K36me3 is instead preventing premature POLII termination. These observations could use more discussion.
6. The numbers of cells analyzed are not always indicated. (eg. EV5). Please check.
7. The authors use g-H2Ax intensity, as opposed to number of foci, for their DNA damage analyses. Why? In our opinion, the intensity approach is more vulnerable to staining artefacts. If the authors wish to rely on an approach using thousands of cells, perhaps they could perform a WB for g-H2Ax instead and quantify?

Answers to reviewers' comments:

Referee #1:

This paper was a pleasure to read. The rationale for each experiment is clearly articulated, the controls for each experiment are present, the approaches are innovative and the conclusions are restricted to what is revealed in the data. Pathways regulating replication timing and the spatio-temporal organization of replication foci are very poorly understood, so this work provides an entryway into one of those pathways. My only struggle with this manuscript came in the last paragraph of the results section. I think we need a few more words to understand why global KD of H3K36me3 leads to a reduction in global H2A.X while local disruption leads to an increase. We have known that aberrations in PCH lead to genome instability for a long time, so that is expected anyhow, but if the authors feel the need to include this final experiment in EV5, the pathway by which global disruption has an apparently opposite effect should be more clearly articulated since all of their other data shows that it will be disrupted at PCH.

Following the recommendation, we have removed the discussion of genome instability upon global loss of H3K36me3, as it is indeed not relevant to the message. We further added the gH2AX level associated with the pericentromeric heterochromatin (PCH) upon targeting the H3.1 K36M to the PCH in Figure EV7B.

Minor comments:

- Something to consider to increase the visibility of this paper: add "through major satellite RNA" to the end of the title.

We have changed the title as recommended.

- When discussing "aberrations" in the RT of heterochromatin, my impression is that these are all RT delays - if correct, why not be more specific and say "leads to delays". Are there cases of advancement of RT?

We have corrected the "aberrations" wherever necessary. We added live-cell time-lapse microscopy analysis (added in Figure 2D & 2E along with Movie EV2 and Movie EV3) to obtain a detailed view of the progression of replication using mRFP-tagged PCNA upon targeting the H3.1 to the PCH. The results show that the replication timing of PCH is delayed, and concomitantly, the replication timing of nuclear lamin-associated domains advances. These overall changes result in a prolonged concomitant replication of PCHs and nuclear/nucleolar-associated domains.

Referee #2:

In this study, Pradhan and colleagues investigated the role of H3K36me3 at pericentric heterochromatin (PCH) in mouse embryonic stem cells (mESCs) during replication. They analyzed the presence of various histone post-translational modifications (PTMs) at PCH throughout the cell cycle and found that H3K36me3 exhibits a dynamic association. To dissect the role of H3K36me3 at PCH, the authors employed several strategies to reduce H3K36me3 levels, including esiRNA knockdown of H3K36 methyltransferases (such as Setd2 and Nsd1). The authors also co-transfect a GFP-tagged histone H3.1 with K-to-M mutation of residue 36 (H3.1K36M) and a GBP-fused MajSat binding protein, thereby targeting the H3K36M to the PCH. The authors argue that these interventions led to defects in replication timing of PCH and altered major satellite (MajSat) expression, particularly affecting S-phase sub-stages SII and SIII. Notably, treatment with locked nucleic acid (LNA) DNA gapmers against the forward but not reverse MajSat transcripts, presumably blocking/removing MajSat transcript(ion) partially mimicked the H3K36me3 reduction phenotype, supporting a model in which H3K36me3 promotes MajSat transcription, which in turn is necessary for timely replication of heterochromatin and maintenance of genome stability. This reviewer has several concerns, related to the underlying mechanism, the explanation of experimental procedures, the clarity of the text, the data visualization in the figures and the data interpretation.

Major concerns:

1) How did the separation of cells based on the replication foci (RFi) occur? Please show several examples in the supplementary. Also, please show the masks and examples of segmented RFi. The same applies to the segmentation of PCH and of other items.

The separation of cells based on the replication foci (RFi) relies on our previous work from Rausch, Weber et al. 2020. The distinct patterns (S I, S II, and S III) were characterized using several parameters, but prominently using live cell microscopy, and RFi spot features (see our answer to point 2). We used the same spatial patterns to distinguish S-phase cells into sub-S phases.

We have added Figure EV2, where we explain how the segmentation and masks are generated in a step-by-step manner. In the same figure, we also have the chromatin compaction classification and mapping of RFi to each class. We have included cell examples with the segmentation masks or classified chromatin.

2) In order to subdivide the S phase cells the authors used the spatial pattern of RFi (EdU) to segregate SI, SII and SIII substages as shown in Fig 1A. How is the spatial pattern calculated to assign a cell to a given group?

The classification of S-phase cells into sub-stages (S I, S II, and S III) is based on previously published quantitative analyses by our group (Rausch, Weber et al., NAR 2020), in which the replication patterns in mouse embryonic stem cells (mESCs) were analyzed using computational, unbiased classification of RFi patterns derived from high-resolution imaging data.

That analysis revealed three robust and reproducible spatial patterns corresponding to different S-phase sub-stages:

S I: RFi distributed diffusely throughout the nucleoplasm, excluding PCH and nuclear/nucleolar periphery.

S II: Characterized by prominent colocalization of RFi with pericentromeric heterochromatin (PCH).

S III: Enrichment of RFi at the nuclear and nucleolar periphery.

For the current study, we manually assigned cells to these stages based on the above-described spatial reference framework, using EdU labeling and chromatin context (DAPI, PCNA, or H3K9me3 as needed). These criteria are well-established in the field and were applied in a consistent manner across all conditions.

To support this manual assignment and expand its validation, we now include:

New time-lapse live-cell microscopy (added in Figure 2D & 2E along with Movie EV2 and Movie EV3) showing transitions through S-phase stages and replication focus remodeling.

Additionally, for conditions such as H3K36me3 depletion (via KMT knockdown or H3.1K36M targeting) and LNA-mediated MaSat RNA knockdown, we assigned cells to the S II sub-stage based on the presence of replication foci colocalizing with PCH-associated structures, specifically: targeted GFP-H3.1 at PCH or high-density DAPI regions.

3) Fig. 1B. The authors introduce seven different chromatin compaction classes. If the reviewer understands correctly, the 1 -> 7 direction corresponds to the grades of chromatin compaction, but this should be explained, e.g. grade 7 corresponds to the most condensed. The basis for this classification is unclear for this reviewer. Please clarify. In the context of the main topic of the manuscript, is the presentation of this data really needed?

We have now clarified the chromatin compaction classification in Figure EV2A, including a schematic and legend that explain the logic behind the seven-class system.

This classification is based on the nucim tool (Schmid et al. 2017), which applies a Hidden Markov Random Field model to voxel-wise DAPI intensity distributions in 3D images. The result is an assignment of each nuclear voxel to one of seven chromatin compaction classes, where: Class 1 corresponds to the least compact (interchromatin) regions, and Class 7 corresponds to the most compacted heterochromatin (e.g., PCHs). The application of this tool allows us to analyze genome organization and replication foci localization in a quantitative and unbiased manner. For ease of understanding, we have assigned classes 1-4 as active and 5-7 as inactive chromatin compartments.

In the context of this study, we believe this analysis is valuable because it provides a second, orthogonal validation of S-phase sub-stage classification. As shown in Figure 1B, each sub-stage (S I, S II, S III) maps onto a distinct chromatin compaction profile, reinforcing the biological relevance of our S-phase staging and the differences in chromatin environments that underlie them.

4) Authors used the EdU pulse chase followed by PCNA staining to monitor the progression of the S phase cells. However, when they assign the cells to an initial S substage based on EdU this should follow the same criteria between a control and treated cell. There are multiple examples where the assignment of a cell based on the EdU pattern (as shown in Fig 1A) may be inappropriate. For example:

In figure EV3 C: the SIII cell of esiKMT has an EdU spatial pattern which resembles the SII pattern.

In figure 2 E: the SII cell of H3.1 K36M has an EdU spatial pattern which resembles the SI pattern.

In figure 4C: the representative EdU patterns for SI and SII look very similar.

If the assignment of a cell to an EdU pattern has been done manually the reviewer is concerned about experimental biases.

Moreover, the applied experimental perturbations may affect replication of some but not other regions of the genome. As a consequence, new combinations of EdU incorporation and PCNA direct labeling patterns may arise. Reviewing the provided images, this reviewer interprets the findings that replication of PCH may be extended, therefore possibly occurring concurrently with replication of perinuclear regions. Hence, an extensive quantification of the data is required, which should be provided in the paper. To understand better a possible selective extension of the duration of replication of PCH, the authors may need to consider applying different durations for the chase phase.

We have replaced the images as applicable. By using the pulse-chase approach, we can get the sequence of the spatial patterns over time. While the G1 to S I or S III to G2 transition is easily recognizable, others remain challenging, especially in the conditions where PCH replication is delayed. Hence, we assign the cells as S II, where the EdU signal has overlap with PCH irrespective of RFI (PCNA) localization in the PCH (e.g., Fig. EV3F esiKMT SII). In the same panel, we assign the cell as S III as it transitioned from late S phase to G2 (absence of bright punctuated PCNA foci). Overall, we found overlap in the S II and S III patterns in the knockdown/H3.1 K36M/ forward MajSat interference conditions, and based on the EdU-PCNA spatial patterns we assigned the cell to S II or S III.

Nonetheless, to get a better time resolution, instead of the different durations of the chase times, live-cell experiments have been performed using targeted H3.1 K36 and H3.1 K36M. We have tracked the cells throughout the entire S-phase with mRFP-PCNA and quantified the duration of PCH replication, as well as the various S-phase substages. With these experiments, we have directly confirmed that PCH replication takes longer in H3.1 K36M targeted cells, together with an earlier start of S III (lamina-associated chromatin replication). Therefore, the concomitant replication of PCH and lamina-associated chromatin (S II + S III) is visualized for a longer period of time in H3.1 K36M targeted cells. (Figure 2D and 2E & Movie EV2 & Movie EV3).

5) The authors used the above type of pulse chase experiments to argue for a delayed cell cycle S phase. Such experiments should provide a relative estimation, and the authors should provide actual quantification of delayed cell cycle in minutes/hours.

The newly added timelapse analysis yields the doubling time and shows delay in the PCH replication start (see also our answer to point 4 above).

6) How do the authors explain the increase of both SETD2 and NSD1 in esiGFP samples from 36 to 48 and further to 72 hours Fig. EV2 B?

As we observed that from 48 hours on the transfection of the control esiGFP led to an increase in the signal from NSD1 and also SETD2, which is not related to the target but the esiRNA transfection alone, we selected the 36 hours time frame as the one with minimal transfection artifacts and maximal downregulation of both SETD2 and NSD1.

Additionally, we note that cells at 36 hours post-transfection were less clustered and more suitable for downstream high-resolution imaging. We selected this timepoint as a compromise that balanced robust knockdown efficiency with optimal sample quality for image analysis. To validate the efficiency, we also performed western blotting.

7) When expressing and recruiting the H3.1K36M protein to the PCH, please show and quantify the accumulation of the GFP anchoring protein at PCH. What do the authors think - does the GFP-tagged histone become incorporated into nucleosomes at PCH, or not? If not, how does the H3K36M protein lower the H3K36me3 levels. Does it locally block the activity of the HMTs? Can one detect an accumulation of the SETD2 or NSD proteins at such PCH regions?

To confirm the recruitment and incorporation of GFP-tagged H3.1 with or without the K36M mutation at the pericentromeric heterochromatin (PCH), we performed fluorescence recovery after photobleaching (FRAP). As a control, we included GFP-MajSat, which is expected to bind dynamically and not incorporate into chromatin. We observed rapid recovery within seconds for GFP-MajSat, consistent with its transient binding behavior. In contrast, both GFP-H3.1-K36 and GFP-H3.1-K36M showed negligible recovery over the observation period, indicating stable incorporation into chromatin at the PCH (Figure EV4C, EV4D and Movie EV1). This supports the notion that the H3.1 variants are incorporated into nucleosomes at the target site, rather than remaining loosely associated.

Although we did not directly detect SETD2 or NSD1 redistribution in this context, the selective and spatially restricted loss of H3K36me3 supports a local inhibition model.

8) The authors argue that forward strand expression of MajSat transcript is somehow involved in the replication of PCH repetitive sequences and that H3K36me3 is acting downstream. To further extend this model, have the authors observed a change in H3K36me3 staining levels upon treatment of the LNA probes against forward but not reverse MajSat transcripts? What is the impact of LNA probe treatment on MajSat forward and reverse transcript levels in the nuclei? Are the LNA probes affecting replication by removing/lowering the local nuclear concentration of MajSat transcripts or could they directly impair the replication process, and only indirectly affect transcript levels? See also below - are majSat transcripts actually transcribed just prior or after their own replication or is it occurring at other stages of the cell cycle?

We have now added the H3K36me3 and H3K9me3 levels in the PCH in S I (prior to S II) (Figure EV6) upon treatment of the LNA-DNA gapmers. We observe the knockdown of reverse affecting the histone modification level more compared to the forward MajSat gapmer treatment, which we have added in the text. We also measured the level of MajSat RNAs after the LNA gapmer treatments by RT-qPCR (Figure EV6A).

While we cannot fully exclude indirect effects of LNA treatment on replication machinery, we believe the observed replication timing changes are primarily due to the loss of MajSat forward transcripts.

Furthermore, the loss of reverse RNA had a more visible impact on H3K36me3 reduction, but did not affect the replication timing, suggesting a direct role of the forward RNA in replication timing regulation.

9) When is H3K36me3 established at PCH during the cell cycle? Does this relate to MajSat expression? Experimentally, could one abrogate expression and/or H3K36me3 deposition prior to replication to see whether then replication of PCH is affected?

Based on our time-resolved imaging and quantitative analysis (Figure 1), we observed that H3K36me3 becomes enriched at pericentromeric heterochromatin (PCH) in G1 and early S phase, peaking in S I, just prior to the replication of PCH in S II. This dynamic enrichment suggests a pre-replication role.

Regarding MajSat RNA expression, our data (Fig. EV5A) demonstrate that forward MajSat transcripts are expressed in S II, overlapping with the replication timing of PCH. We also observe notable expression post S II for MajSat forward, although the expression goes down in G2. Thus, H3K36me3 deposition precedes and potentially facilitates transcription of MajSat RNA, which in turn influences replication timing, as supported by the knockdown experiments (Figure 4).

Reducing the H3K36me3 level before S II remains challenging, as known lysine demethylases (KDMs; for example, JMJD2B/KDM4B) for H3K36me3 are also known to demethylate H3K9me3; hence, targeting the KDMs to PCH would also lead to the loss of H3K9me3.

10) The authors should provide a proper discussion addressing a) the role of H3K36 demethylases at PCH, given that at least KDM2A was shown to interact with HP1, b) the role of classic PCH histone PTMs such as H3K9me3 and H4K20me3 along with their corresponding methyltransferases (Suv39h1/2, Setdb1, Suv420h1/2) in replication, c) the effect at other heterochromatic components which might overlap with DAPI bright foci, such as telomeric repeats, as well as the d) the similarities and differences of the esiKMT and H3.1K36M PCH targeting experiments.

We have added these points to the discussion.

Minor Comments:

1. The authors show that relative percentages of cells in G1 S and G2 are changed between esiGFP and esiKMT which raises the question of the relative % of S I, S II and S III population change upon knockdown of KMT or expression of H3.1 K36M or knockdown of maj sat RNA? Please specify.

While we fully agree that quantifying the relative proportions of S I, S II, and S III substages across all experimental conditions would provide additional insight, such an analysis was technically challenging for certain perturbations due to variability in transfection efficiency and the asynchronous nature of the cell population. However, we were able to perform live-cell imaging and quantification of S-phase progression dynamics specifically for the H3.1-K36 vs. H3.1-K36M targeting experiment, where mRFP-tagged PCNA enabled reliable replication tracking.

2. DAPI preferentially binds to AT-rich DNA stretches, present in major satellite repeat sequences. These sequences form the DAPI bright chromocenter foci which show dynamic changes in numbers and sizes during cell cycle and development. Can these changes affect the measured intensity of the DAPI signal and affect the applied normalizations in some way?

Indeed, DAPI preferentially binds AT-rich DNA, such as major satellite repeats that form the pericentromeric heterochromatin (PCH). These DAPI-bright chromocenters do show dynamic structural changes during the cell cycle and development.

To minimize the impact of such variability on our measurements, we implemented the following safeguards:

- DNA Content Normalization: In quantitative analyses (e.g., for histone modification levels or EdU incorporation), signal intensities were normalized to total DAPI intensity per nucleus, which serves as a proxy for DNA content and mitigates variability across individual nuclei.

- Voxel-Based Segmentation: For spatial analysis, we used the nucim tool with a hidden Markov model to classify chromatin compaction classes based on DAPI intensity, which takes local intensity context into account and is less sensitive to global variation in DAPI brightness.

- Manual Validation of PCH Segmentation: For measurements specifically within PCH, regions were segmented based on a thresholding method consistent across conditions and manually validated using DAPI intensity and co-localization with H3K9me3 or GFP-MaSat when applicable.

3. Can the knockdown of MajSat transcripts be verified by FISH or PCR?

We verified using qPCR (see Figure EV6 A).

4. Is there a dynamic expression of MajSat transcripts during cell cycle and S phase substages? Are the MajSat transcripts RNA expression levels affected by KMT knockdown or expression of H3.1 K36M show a cell cycle specific manner?

We characterized the cell cycle-dependent dynamic expression and added to the Figure EV5.

5. In figure EV5B, are the micronuclei observed in a particular cell cycle stage or substage of the S phase?

In the current study, we quantified micronuclei formation as a general proxy for genome instability across populations but did not specifically assess the cell cycle stage or sub-S-phase classification of the cells harboring micronuclei in Figure EV7D.

Future work with live-cell imaging or multiplexed cell cycle markers could help address whether micronuclei formation is enriched in a specific cell cycle stage, particularly following replication timing defects.

We now acknowledge this limitation in the revised manuscript text and discussion.

6. Fig. EV3C: Since the analysis of chromatin was based on DAPI staining, please show the DAPI channels accordingly in the same panel. The t1 and t2 labels in this and similar other panels can be removed, as from the description of the pulse and chase method (Methods section), the timing of EdU incorporation and chase duration are clear. This is relevant to other panels organized in the same principle.

The DAPI channel is included in the merged images, which were used for both visualization and chromatin segmentation.

7. Results and Discussion, Part 1, Paragraph 1: "However, most of the epigenetic states are re-established, as inferred from the normalized PTM levels in G2, except for H3K9me2, H3K36me3, H4K8ac and H4K12ac (summarized in Fig. EV1C)" : this argument is an oversimplification since only 11 out of 13 PTM show significant changes in G2 compared to G1. Thus, there is only a minority of PTMs which fully reestablish their modification levels after the S phase. Please correct.

We have corrected this in the text now.

8. Please explain the rationale for the measurement of MajSat RNA in different cell cultures?

We have now removed these results as it is not highly relevant in this context.

9. It would be a bit easier for the reader if the authors would briefly explain what the advantage of using asynchronous cell culture is, instead of the synchronized, in pulse chasing experiments. I guess, this is to capture all different cells.

We have now described the advantages in the text. Asynchronous cell cultures ensure that all cell cycle stages are represented simultaneously and without any (chemical) perturbations to the cell cycle, as many commonly used drugs are also known to affect chromatin to a certain extent.

10. Given that knockdowns of SETD1 and NSD1 affect H3K36me3 genome wide and part of H3.1 K36M is not at PCH (although the majority is properly targeted), the authors should discuss indirect effects of PCH replication timing effect or in general replication related effects.

Indeed, both the esiRNA-mediated knockdown of SETD2 and NSD1 and the H3.1 K36M targeting approach may introduce indirect effects due to their broader impact on H3K36me3 levels outside of pericentromeric heterochromatin (PCH). While the nanotrap-based targeting efficiently enriched H3.1-K36M at PCH, some degree of incorporation at non-PCH regions cannot be ruled out.

We also emphasize that the use of strand-specific MajSat RNA knockdown, which acts independently of histone perturbations, produced a replication phenotype similar to H3K36me3 loss at PCH. This strengthens our conclusion that the effects observed are at least partially direct and PCH-specific.

11. Results and discussion, paragraph 2, sentence: "based on the spatial pattern of the RFi" - please specify what are the RFi already in this sentence, as this abbreviation has not been introduced before .

We have now added the abbreviation.

12. In this sentence: "The tandem repeats of MajSat comprising the PCH are generally transcriptionally incompetent, but these regions are known to potentially be transcribed", the phrasing is contradictory and not in line with the literature.

We have now corrected the text.

13. There are two figures termed as EV4. One is related to the transcription of MajSat in cell lines, and another is about Pol II Ser2 staining upon targeting of H3.1K36M to the PCH. Please correct the labels accordingly in figure legends and in the text.

We have now corrected the figure legends.

14. Fig. 2D and Fig EV3B are redundant- the same cartoon appears in both.

We have now removed the illustration.

15. Fig. 1D. Please report n for each individual stage

We have now added the n.

16 . The authors should cite literature on why they use gH2A.X levels as a proxy of genome instability.

We have now cited these articles (Rogakou et al., 1999; Sedelnikova et al., 2002).

17. Please provide consistent magnifications of the figures for the gH2A.X immunofluorescence data in esiGFP/esiKMT and the H3K36M experiments in figure EV5 A-C.

We have now corrected this part.

18. The description of the gH2A.X antibody is not available in the materials and methods, please update the table accordingly.

We have now added the antibody information. Thank you for pointing that out.

19. Fig. EV3 B - in the cartoon, why do the red dots disappear in t2? If they correspond to a short pulse of EdU incorporated, they should still be present. According to the cartoon, it may seem that the incorporation of EdU was done for 3 hours, while it was only for 10 mins. Please make the cartoon clearer about the timing of incorporation of EdU, thymidine, and the subsequent fixation of cells.

We have now corrected this part.

20. Many sentences can be improved from the perspective of clarity and typos, especially those in figure legends. Examples as follows:

"Briefly, a short pulse of EdU was followed by a chase with an excess thymidine to block further EdU incorporation for three hours and fixed" - please specify what was fixed.

Fig. 1E,F - please correct the legend, as panels E and F are labelled as B and C, accordingly

"Replication timing also influences the epigenome and plays a key role in maintaining the same." Please clarify what is the same?

We have now clarified this.

Referee #3:

In their manuscript titled "Dynamic association of H3K36me3 with pericentromeric heterochromatin promotes its transcription to regulate replication timing" the authors Pradhan et al. delve into understanding the role of the histone post translational modification H3K36me3 in DNA replication and satellite DNA transcription. Specifically, by using asynchronously growing mouse embryonic stem cells, they discovered that H3K36me3 is dynamically regulated at different phases of the cell cycle. In particular, H3K36me3 is upregulated during the G1/S transition and co-localizes with pericentromeric heterochromatin (PCH) prior to the replication of this domain. To test whether H3K36me3 has a role in regulating DNA replication at the PCH, they knocked down the major lysine methyltransferases responsible for H3K36me3 modification to generally reduce H3K36me3. In a parallel approach, they used a sophisticated strategy to specifically target H3K36me3 only at pericentromeric regions using forced integration of a mutated version of H3K36 that cannot be methylated. In both experiments, they observed alteration of chromatin replication timing, indicating that H3K36me3 may play a role in regulating this process. In addition, they also discovered that upon loss of H3K36me3, the non-coding forward transcript of MajSatDNA is dramatically reduced. By selectively impairing the function of forward MajSatRNA, the authors found similar results to the loss of H3K36me3 - altered replication timing. The authors propose a final speculative model in which H3K36me3 is important for MajSat DNA transcription, with the satellite DNA transcript in turn relevant for chromatin replication timing. The data presented in this manuscript is of good quality, relevant and mostly well-interpreted.

In our opinion, the major conceptual advance in this manuscript is that H3K36 trimethylation is associated MajSat DNA transcription and DNA replication timing. However, we think that other aspects of the manuscript remain underdeveloped, particularly the question of if and how MajSat influences replication timing. In addition, some experiments remain underquantified (see below) while certain conclusions are also not completely supported by data.

Major comments:

1. We found the nuclear compartment classification system (INC/ANC, IC/PR/CDC) very confusing and a schematic is absolutely required in Fig. 1.

We have added a schematic in Figure EV2. We have also introduced an explanatory scheme under the nuclear compartment classification figures.

2. In general, the microscopy images of the KMT depletion are not convincing, but consistent with the quantifications in EV2B. What was the rationale to not use samples at 48H when the fold change between eGFP control and KMT KD is higher?

As we observed that from 48 hours on the transfection of the control esiGFP led to an increase in the signal from NSD1 and also SETD2, which is not related to the target but the esiRNA transfection alone, we selected the 36 hours time frame as the one with minimal transfection artifacts and maximal downregulation of both SETD2 and NSD1.

Additionally, we note that cells at 36 hours post-transfection were less clustered and more suitable for downstream high-resolution imaging. We selected this timepoint as a compromise that balanced robust knockdown efficiency with optimal sample quality for image analysis. To validate the efficiency, we also performed the Western blot.

3. In the pulse-chase approach to quantify replication timing, the authors use excess thymidine to block further EdU incorporation. Why does replication proceed in the presence of excess thymidine? In general, how this pulse chase experiment works could be explained better.

In our pulse-chase setup, we use excess thymidine (50 μ M for 10 minutes) to compete out EdU (10 μ M for 10 minutes) after a short EdU pulse. This concentration is sufficient to block further EdU incorporation by mass action without impairing DNA replication, as thymidine is a natural nucleotide. Thus, replication proceeds normally, but no new EdU is incorporated during the chase. After this, we remove and replace with fresh media. We have now described this part in more detail in the methods section.

4. In the experiment where the onco-histone H3.1 K36R is targeted to the MajSat DNA, are H3K36 methylation levels only specifically reduced at the PCH? Do the authors observe that H3K36 methylation at other loci is unaffected?

To confirm the recruitment and incorporation of GFP-tagged H3.1 with or without the K36M mutation at pericentromeric heterochromatin (PCH), we performed fluorescence recovery after photobleaching (FRAP). As a control, we used GFP-MajSat, which is expected to bind dynamically and not incorporate into chromatin. GFP-MajSat exhibited rapid fluorescence recovery within seconds, consistent with transient binding. In contrast, both GFP-H3.1-K36 and GFP-H3.1-K36M showed negligible recovery over the observation period, indicating stable incorporation into chromatin at the PCH (Figure EV4C, EV4D).

Although we did not directly quantify global H3K36me3 levels in this targeted condition, our microscopy-based readouts show that the reduction of H3K36me3 is specifically observed at the PCH, while other nuclear regions show no apparent loss in signal intensity. Thus, our data support a localized reduction of H3K36me3 at PCH following targeted expression of H3.1-K36M. Of course, we do not rule out minor fractions of H3.1 to be incorporated into chromatin outside PCH.

5. In general, the replication timing data in EV3C, 2E & 4 is not quantified. The role of H3K36me3/MajSat transcription in replication timing is a central conclusion of the manuscript but is only supported by representative images.

We have now added the quantification.

6. In Figure 3 (and referred to on page 7), the authors used RNA pol II Ser5 staining to visualize active transcription sites. However, these images are missing from Figure 3, and they need to be shown. In addition, the authors do not fully discuss the implications of this somewhat counterintuitive result.

We have now added the images and pointed out the PCH in the zoomed version of the images.

7. On page 7-8, the authors titled their last result section as, 'H3K36me3-dependent expression of MajSat (forward) maintains the replication timing of constitutive heterochromatin and genome stability'. However, we think that there is not enough evidence to show an active role of MajSat RNA in maintaining chromatin replication timing. First, the gapmers against MatSat RNA have not been validated and whether the knockdown is effective remains unknown. Also, the authors conclude that it is the loss of MajSat forward transcripts that leads to chromatin replication time disruption. However, this could be due to other effects associated with loss of H3K36 methylation. To make their conclusion stronger, the authors could try to perform a rescue experiment in which they introduce forward MajSatRNA in a background in which H3K36me3 is reduced, and see whether this rescues chromatin replication timing. Alternatively, they should interpret their data more carefully.

We validated the expression level of MajSat RNA after knockdown, and also quantified their effects on both H3K36me3 and H3K9me3 levels inside PCH (Figure EV6). Interestingly, the reverse MajSat interference affected the histone PTM level more than

the forward MajSat interference, where the level was comparable to control, but certainly both conditions affected the epigenetic nature of PCH. This suggests the MajSat forward is independently regulating the replication timing.

The experiment suggested, albeit very interesting, it is quite complex as to perform a controlled and time-dependent expression of MajSat forward RNA is unlikely to work efficiently, and in addition the level should be tunable to match endogenous expression concomitant with reducing the level of H3K36me3 also locally at major satellite chromatin. We have, thus, interpreted our data more carefully and pointed out limitations.

Minor:

1. There are several instances in the manuscript where the choice of words are unconventional, and therefore imprecise. I've listed two examples below but the manuscript could use copy editing overall.

a. Page 3 - '...significant disruption in histone PTM levels....'. Should be 'reduction/depletion' etc.

b. Page 4 - '....known to be transcriptionally incompetent,....' Should be 'silent'

We have now corrected this contradictory sentence in the text.

2. Abbreviations - RFI is used in page 3 but without indicating what it stands for. Similarly, esiRNA (page 5) should be properly described.

We have now corrected this in the text.

3. On page 5, the authors use H3K9me3 as a negative control for their GFP nano-trap experiment to verify that another H3 post-translational modification is not affected. This should be better explained in the text.

We have now discussed this in text.

4. Figure 2 B, C, p-values and statistical analysis are missing.

We have now added statistical analysis.

5. On page 7, the authors speculate that H3K36me3 could prevent premature transcription initiation (perhaps by keeping POLII paused?) and promoting elongation. Another possibility is that H3K36me3 is instead preventing premature POLII termination. These observations could use more discussion.

We have now added this to the discussion.

6. The numbers of cells analyzed are not always indicated. (eg. EV5). Please check.

We have now added the n in all figures.

7. The authors use g-H2Ax intensity, as opposed to number of foci, for their DNA damage analyses. Why? In our opinion, the intensity approach is more vulnerable to staining artefacts. If the authors wish to rely on an approach using thousands of cells, perhaps they could perform a WB for g-H2Ax instead and quantify?

We have now removed the effect of global knockdown of KMTs (SETD2+NSD1) as it was not relevant in this particular context. For the targeting of PCH experiments, we select the cells based on their target efficiency visually, as not all cells show this targeting efficiently, making this difficult for western blotting analysis.

Dear Prof. Cardoso,

Thank you for the submission of your revised manuscript to our editorial offices. I have now received the reports from the three referees that I asked to re-evaluate the study, you will find below. As you will see, the referees now support the publication of your study in EMBO reports. However, referees #2 and #3 have remaining concerns or further comments and suggestions to improve the manuscript, I ask you to address in a final revised manuscript. Please also provide a final p-b-p-response to these points and my editorial requests below.

Editorial requests:

- Please provide a final manuscript title with not more than 100 characters including spaces.
- Please provide the abstract written in present tense throughout.
- Please clearly mark the corresponding author on the title page and add an e-mail contact.
- We plan to publish your manuscript as report, as also selected by you during submission. Thus, we need 5 final main figures and 5 final EV figures. I would suggest to combine some of the present EV figures to have 5 final ones. Please then update their legends and all the affected callouts.
- Please make sure all figure panels (main and EV figures) are called out separately and sequentially. Presently, there seem to be no separate callouts for panels 2B and 2C. Please check.
- We now use CRediT to specify the contributions of each author in the journal submission system. CRediT replaces the author contribution section. Please use the free text box to provide more detailed descriptions and do NOT provide your final manuscript text file with an author contributions section. See also our guide to authors: <https://www.embopress.org/page/journal/14693178/authorguide#authorshipguidelines>
- Some of the scale bars in the microscopic images are rather thin and hard to see. Please improve.
- Please check again that the number "n" for how many independent experiments were performed, their nature (biological versus technical replicates), the bars and error bars (e.g. SEM, SD) and the test used to calculate p-values is indicated in the respective figure legends. Please also check that all the p-values are explained in the legend, and that these fit to those shown in the figure. Please provide statistical testing where applicable. Please avoid the phrase 'independent experiment' but clearly state if these were biological or technical replicates. Please also indicate (e.g. with n.s.) if testing was performed, but the differences are not significant. In case n=2, please show the data as separate datapoints without error bars and statistics. See also: <http://www.embopress.org/page/journal/14693178/authorguide#statisticalanalysis>

If n<5, please show single datapoints for diagrams. Moreover:

- Please define the annotated p values ****/****/**/* as well as provide the exact p-values for the same in the legend of figure 4D as appropriate.
- Please note that the exact p values are not provided in the legends of figures 2C, 3B, C, D, E; EV3 B; EV4 F, EV5A, B; EV6 A.
- Please indicate the statistical test used for data analysis in the legends of figures 3B, E.
- Please note that the box plots need to be defined in terms of minima, maxima, centre, bounds of box and whiskers, and percentile in the legends of figures EV1 A
- Please note that information related to n is missing in the legends of figures 4D, EV1 A
- Please note that the error bars are not defined in the legends of figures 3B, EV6 A.

- Please add to each legend (main, EV figures and Appendix Figures, where applicable) a 'Data Information' section (or name the provided 'notes' section like this) explaining the statistics used or providing information regarding replicates and scales. See:

- The data availability section (DAS) is restricted to externally deposited large datasets generated in a study. If no primary datasets have been deposited, please state this in this section ("No large primary datasets have been generated and deposited for this study"). Please remove all other information from this section.

- Presently some of the microscopic images overlap or are placed close to each other and it seems they are one image (e.g. Fig. 1A merged, or Fig. 2D). Please separate different microscopic images in the figures clearly by white or black lines, depending on the background.

- Please provide legends for the movies. Each legend should be provided as a readme.txt file and then should be zipped up with its corresponding movie file so that we have 3 ZIP folders uploaded as Movie EV1-Ev3.

- All Materials and Methods need to be described in the main text using our 'Structured Methods' format, which is required for all research articles. According to this format, the Methods section should include a Reagents and Tools Table (listing key reagents, experimental models, software, and relevant equipment and including their sources and relevant identifiers), uploaded as separate file, and a Methods section in which we encourage the authors to describe their methods using a step-by-step protocol format with bullet points, to facilitate the adoption of the methodologies across labs. More information on how to adhere to this format as well as downloadable templates (.doc) for the Reagents and Tools Table can be found in our author guidelines (section 'Structured Methods'):

- Please add the information provided in Table 1 and Table 2 to the Reagents and Tools Table (and remove them from the manuscript text file). Moreover, please add other primer information provided throughout the methods section to the Reagents and Tools table (and remove this from the text). Please add callouts to the R & T table were applicable.

- Thanks for providing the source data. However, please upload the source data directly to the submission system (as indicated in the source data information sent to you previously) as one folder per main figure, grouping together all the files for this figure (and ZIPed together), and as one folder for the EV figures containing separate folders for each EV figure).

In addition, I would need from you uploaded separately:

Please use this link to submit your revision: <https://embor.msubmit.net/cgi-bin/main.plex>

Best,

Referee #1:

As is evident from my first review, I thought this paper was interesting and one of only few in my 30 years doing this that I thought was ready for publication with just a few clarifications, which the authors have now taken care of. I don't want to take anything away from the other reviewers' critiques - it is possible that I did not have many of the same problems as they did because I have followed the Cardoso lab from the beginning and am very familiar with all the tools. It is important for them to address the other reviewers' comments because other readers my experience similar confusion. To me, this was great example of the Cardoso lab depth of thinking and approach to the replication problem. Beautiful work!!!

Referee #2:

This reviewer acknowledges the additional experiments done by the authors which improved the quantitative description of the findings as well as provided a better molecular understanding of some of the observations reported previously.

However, this reviewer thinks that the molecular links between an H3K36me3 pathway and major satellite transcripts/transcription are not clear cut and are not well resolved in the paper. To me, the paper shows that H3K36 methylation disturbance impacts on replication timing of pericentromeric heterochromatin within chromocenters. Likewise, expression of LNA-gapmers against forward major satellites impairs replication timing of chromocenters (Fig 4). However, the later perturbation does not have a major impact on H3K36me3 levels (EV6). In contrast, while expression of LNA-gapmers against reverse major satellites does not impair replication timing of chromocenters (Fig. 4), it leads to a major reduction of H3K36me3 levels (EV6).

Hence, the title of the paragraph on page 8 ("H3K36me3-dependent expression of MajSat (forward) RNA maintains the replication timing of constitutive heterochromatin and genome stability) is not supported by the data currently provided.

The molecular mechanism linking H3K36me3 to replication via satellite transcription is currently not resolved. The authors are urged to more thoroughly and clearly discuss what conclusions can be drawn from the current data and which aspects still need to be further investigated in more molecular detail.

Finally, following this line of arguments, the revised title of the paper should be critically reconsidered. The addition of "through major satellite RNA" is not (yet) supported by the data. There is some molecular interconnection, but the paper does not provide a clear mechanism linking all observations together.

Minor point:

To facilitate a better understanding of how the authors specified cells to be within a SII or SIII state, this reviewer suggests including part of the rebuttal response into the main manuscript as follows:

From rebuttal page 3, point 4:

"Overall, we found overlap in the S II and S III patterns in the knockdown/H3.1 K36M/ forward MajSat interference conditions, and based on the EdU-PCNA spatial patterns we assigned the cell to S II or S III."

Part of the sentence "and based on the EdU-PCNA spatial patterns we assigned the cell to S II or S III." is critical. Basically, as I understand it, the authors scored cells to be in SIII when the labeling for PCNA is not detectable (anymore).

This notion could somehow, for example, be incorporated in the main manuscript text on page 5 within the following paragraph: "In controls (esiGFP treated cells), we did not observe any deviation of replication progression, and it was similar to the earlier observed replication program in mESCs, where euchromatin replication in S I was followed by the PCH in the S II, and nuclear periphery and nucleolar associated chromatin replicated in S III (Rausch et al, 2020). Following knockdown of KMTs (esiKMT), we observed an aberration in the progression of the heterochromatin replication timing (Fig. EV3F). After the euchromatin replication in S I, we observed a concomitant replication of PCH with nuclear periphery and nucleolar-associated chromatin, whose replication timing is usually temporally separated."

Referee #3:

The authors have generally improved their manuscript, including quantifications for many figures. What still remains unresolved, in my opinion, is the data on the Maj Sat RNA. In EV6A, the authors attempt to validate the KD but do not see a significant difference for the forward strand. In addition, the magnitude of the depletion (if indeed depleted) is below 20%. Unfortunately, it is this condition where they report a change in replication timing in Fig 4C-D. Am I missing something? If not, the fact that they have emphasized the role of Maj Sat RNA in the title of the revised manuscript makes this an important point to resolve.

Answers to comments/requests:

Editorial requests:

- Please provide a final manuscript title with not more than 100 characters including spaces.

We have changed the title as required.

- Please provide the abstract written in present tense throughout.

We have changed to the present tense as required.

- Please clearly mark the corresponding author on the title page and add an e-mail contact.

We have added the details.

- We plan to publish your manuscript as report, as also selected by you during submission. Thus, we need 5 final main figures and 5 final EV figures. I would suggest to combine some of the present EV figures to have 5 final ones. Please then update their legends and all the affected callouts.

We combined the figures EV5, 6, and 7 into a new figure EV5 and changed the corresponding figure legends and citations. We have also removed Figure 5 (summary figure) and presented it as a synopsis.

- Please make sure all figure panels (main and EV figures) are called out separately and sequentially. Presently, there seem to be no separate callouts for panels 2B and 2C. Please check.

Done!

- We now use CRediT to specify the contributions of each author in the journal submission system. CRediT replaces the author contribution section. Please use the free text box to provide more detailed descriptions, and do NOT provide your final manuscript text file with an author contributions section. See also our guide to authors:

<https://www.embopress.org/page/journal/14693178/authorguide#authorshipguidelines>

We have removed the author contribution section from the manuscript text file.

- Some of the scale bars in the microscopic images are rather thin and hard to see. Please improve.

We have changed the thickness of the scale bars.

- Please check again that the number "n" for how many independent experiments were performed, their nature (biological versus technical replicates), the bars and error bars (e.g. SEM, SD) and the test used to calculate p-values is indicated in the respective figure legends. Please also check that all the p-values are explained in the legend, and that these fit to those shown in the figure. Please provide statistical testing where applicable. Please avoid the phrase 'independent experiment' but clearly state if these were biological or technical replicates. Please also indicate (e.g., with n.s.) if testing was performed, but the differences are not significant. In case n=2, please show the data as separate datapoints without error bars and statistics. See also:

<http://www.embopress.org/page/journal/14693178/authorguide#statisticalanalysis>

If n<5, please show single datapoints for diagrams. Moreover:

- Please define the annotated p values ****/***/**/* as well as provide the exact p-values for the same in the legend of Figure 4D as appropriate.

- Please note that the exact p values are not provided in the legends of figures 2C, 3B, C, D, E; EV3 B; EV4 F, EV5A, B; EV6 A.

- Please indicate the statistical test used for data analysis in the legends of figures 3B, E.

- Please note that the box plots need to be defined in terms of minima, maxima, centre, bounds of box and whiskers, and percentile in the legends of figures EV1 A
- Please note that information related to n is missing in the legends of figures 4D, EV1 A
- Please note that the error bars are not defined in the legends of figures 3B, EV6 A.

We have changed and added the required information to the figure legends

- Please add to each legend (main, EV figures and Appendix Figures, where applicable) a 'Data Information' section (or name the provided 'notes' section like this) explaining the statistics used or providing information regarding replicates and scales. See:

We have updated the data information section.

- The data availability section (DAS) is restricted to externally deposited large datasets generated in a study. If no primary datasets have been deposited, please state this in this section ("No large primary datasets have been generated and deposited for this study"). Please remove all other information from this section.

We have kept the DAS section as there are large, microscopy datasets, which we can only provide via our institutional repository.

- Presently some of the microscopic images overlap or are placed close to each other and it seems they are one image (e.g. Fig. 1A merged, or Fig. 2D). Please separate different microscopic images in the figures clearly by white or black lines, depending on the background.

We have separated the merged images for better visualization.

- Please provide legends for the movies. Each legend should be provided as a readme.txt file and then should be zipped up with its corresponding movie file so that we have 3 ZIP folders uploaded as Movie EV1-Ev3.

We have added the legends and movies in the zipped folders.

- All Materials and Methods need to be described in the main text using our 'Structured Methods' format, which is required for all research articles. According to this format, the Methods section should include a Reagents and Tools Table (listing key reagents, experimental models, software, and relevant equipment and including their sources and relevant identifiers), uploaded as separate file, and a Methods section in which we encourage the authors to describe their methods using a step-by-step protocol format with bullet points, to facilitate the adoption of the methodologies across labs. More information on how to adhere to this format as well as downloadable templates (.doc) for the Reagents and Tools Table can be found in our author guidelines (section 'Structured Methods'):

- Please add the information provided in Table 1 and Table 2 to the Reagents and Tools Table (and remove them from the manuscript text file). Moreover, please add other primer information provided throughout the methods section to the Reagents and Tools table (and remove this from the text). Please add callouts to the R & T table where applicable.

We have made the table and changed the callouts.

- Thanks for providing the source data. However, please upload the source data directly to the submission system (as indicated in the source data information sent to you previously) as one folder per main figure, grouping together all the files for this figure (and ZIPed together), and as one folder for the EV figures containing separate folders for each EV figure).

We have prepared two zip folders.

In addition, I would need from you uploaded separately:

- a short, two-sentence summary of the manuscript (not more than 35 words).

- two to four short (!) bullet points highlighting the key findings of your study (two lines each).
- a schematic summary figure as separate file that provides a sketch of the major findings (not a data image) in jpeg or tiff format (with the exact width of 550 pixels and a height of not more than 400 pixels) that can be used as a visual synopsis on our website.

We have prepared and included all the above.

All changes to the text are labeled in red font.

Referee #1:

As is evident from my first review, I thought this paper was interesting and one of only few in my 30 years doing this that I thought was ready for publication with just a few clarifications, which the authors have now taken care of. I don't want to take anything away from the other reviewers' critiques - it is possible that I did not have many of the same problems as they did because I have followed the Cardoso lab from the beginning and am very familiar with all the tools. It is important for them to address the other reviewers' comments because other readers my experience similar confusion. To me, this was great example of the Cardoso lab depth of thinking and approach to the replication problem. Beautiful work!!!

We thank the reviewer for the very positive assessment of our work and for recognizing the depth of our approach to replication timing. We appreciate the reviewer's perspective and agree that clarifying our methodology and interpretations is essential to make the study accessible to a broad readership.

Referee #2:

This reviewer acknowledges the additional experiments done by the authors which improved the quantitative description of the findings as well as provided a better molecular understanding of some of the observations reported previously.

However, this reviewer thinks that the molecular links between an H3K36me3 pathway and major satellite transcripts/transcription are not clear cut and are not well resolved in the paper. To me, the paper shows that H3K36 methylation disturbance impacts on replication timing of pericentromeric heterochromatin within chromocenters. Likewise, expression of LNA-gapmers against forward major satellites impairs replication timing of chromocenters (Fig 4). However, the later perturbation does not have a major impact on H3K36me3 levels (EV6). In contrast, while expression of LNA-gapmers against reverse major satellites does not impair replication timing of chromocenters (Fig. 4), it leads to a major reduction of H3K36me3 levels (EV6).

Hence, the title of the paragraph on page 8 ("H3K36me3-dependent expression of MajSat (forward) RNA maintains the replication timing of constitutive heterochromatin and genome stability) is not supported by the data currently provided.

We thank the reviewer for this important point. We agree that our data do not yet establish a direct mechanistic link between H3K36me3 and both MajSat RNAs in regulating replication timing. Instead, our results suggest the loss of H3K36me3 or MajSat forward RNA represents sequential but partly separable pathways influencing PCH replication dynamics.

Forward MajSat RNA interference impaired replication timing without significantly reducing H3K36me3, whereas reverse MajSat RNA interference reduced H3K36me3 without markedly altering replication timing. Notably, H3K9me3 also reduced significantly concomitant with H3K36me3 reduction upon MajSat reverse RNA interference. These observations highlight a complex regulatory interplay between histone modification and strand-specific satellite transcription, which requires further mechanistic dissection in future work. We have accordingly changed the section title and revised the Discussion to clarify the conclusions that can be drawn from our present data. We also revised the main manuscript title to better reflect the current evidence.

The molecular mechanism linking H3K36me3 to replication via satellite transcription is currently not resolved. The authors are urged to more thoroughly and clearly discuss what conclusions can be drawn from the current data and which aspects still need to be further investigated in more molecular detail.

Finally, following this line of arguments, the revised title of the paper should be critically reconsidered. The addition of "through major satellite RNA" is not (yet) supported by the data. There is some molecular interconnection, but the paper does not provide a clear mechanism linking all observations together.

We agree that the direct molecular mechanism linking H3K36me3 to replication timing via major satellite RNA is not yet fully resolved. Our current data strongly support a sequential model in which H3K36me3 enrichment at pericentromeric heterochromatin promotes major satellite transcription, but only the forward-strand RNA contributes directly to replication timing, while the reverse-strand RNA primarily affects heterochromatin marks without altering replication timing. We have revised the text in the Results and Discussion to clarify this interpretation, and we now emphasize that our findings uncover a correlation and functional contribution of forward-strand MajSat RNA, but further work will be required to resolve the molecular mechanism.

Minor point:

To facilitate a better understanding of how the authors specified cells to be within a SII or SIII state, this reviewer suggests including part of the rebuttal response into the main manuscript as follows:

From rebuttal page 3, point 4:

"Overall, we found overlap in the S II and S III patterns in the knockdown/H3.1 K36M/forward MajSat interference conditions, and based on the EdU-PCNA spatial patterns we assigned the cell to S II or S III."

Part of the sentence "and based on the EdU-PCNA spatial patterns we assigned the cell to S II or S III." is critical. Basically, as I understand it, the authors scored cells to be in SIII when the labeling for PCNA is not detectable (anymore).

This notion could somehow, for example, be incorporated in the main manuscript text on page 5 within the following paragraph:

"In controls (esiGFP treated cells), we did not observe any deviation of replication progression, and it was similar to the earlier observed replication program in mESCs, where euchromatin replication in S I was followed by the PCH in the S II, and nuclear periphery and nucleolar associated chromatin replicated in S III (Rausch et al, 2020). Following knockdown of KMTs (esiKMT), we observed an aberration in the progression of the heterochromatin replication timing (Fig. EV3F). After the euchromatin replication in S I, we observed a

concomitant replication of PCH with nuclear periphery and nucleolar-associated chromatin, whose replication timing is usually temporally separated."

We thank the reviewer for pointing this out. As suggested, we have now clarified in the Results section how cells were classified into S I, SII or SIII and have incorporated into the text and corresponding figure legend in EV 3.

Referee #3:

The authors have generally improved their manuscript, including quantifications for many figures. What still remains unresolved, in my opinion, is the data on the Maj Sat RNA. In EV6A, the authors attempt to validate the KD but do not see a significant difference for the forward strand. In addition, the magnitude of the depletion (if indeed depleted) is below 20%. Unfortunately, it is this condition where they report a change in replication timing in Fig 4C-D. Am I missing something? If not, the fact that they have emphasized the role of Maj Sat RNA in the title of the revised manuscript makes this an important point to resolve.

We agree that the observed depletion of forward MajSat RNA upon LNA-gapmer interference appears modest when quantified by qPCR (EV6A now EV5). We note, however, that repetitive elements such as major satellites are notoriously difficult to target and quantify due to their repetitive nature, copy number, and sequence redundancy. Even small changes in transcript levels can reflect substantial local effects on chromatin organization, especially if forward-strand RNAs act in cis at pericentromeric heterochromatin.

In line with this, despite the modest average depletion detected by qPCR, we consistently observed significant replication timing defects (Fig. 4C–D), phenocopying the loss of H3K36me3. This suggests that replication timing is highly sensitive to forward MajSat RNA levels and that even partial reduction is sufficient to disrupt proper progression.

We have now clarified this point in the Results and Discussion by explicitly stating the limited depletion levels, the challenges of measuring repetitive RNAs, and by tempering our interpretation to emphasize that forward MajSat RNA contributes to replication timing, but the precise quantitative threshold and mechanism remain to be determined.

Prof. M. Cristina Cardoso
TU Darmstadt
Biology
Schnittspahnstr. 10
Darmstadt 64287
Germany

Dear Prof. Cardoso,

Thank you for the submission of your final revised manuscript to our editorial offices. I now went through it and your final p-b-p-response and I consider the remaining concerns and requests of the referees as adequately addressed.

I am thus very pleased to accept your manuscript for publication in the next available issue of EMBO reports. Thank you for your contribution to our journal.

Yours sincerely,
